# Impaired primitive erythropoiesis and defective vascular development in Trim71-KO embryos

Tobias Beckröge[1], Bettina Jux[1], Hannah Seifert[1], Hannah Theobald[2], Elena De Domenico[3,4,5], Stefan Paulusch[4,5], Marc Beyer[4,5,6], Andreas Schlitzer[2], Elvira Mass[7], Waldemar Kolanus[1]

**The transition of an embryo from gastrulation to organogenesis requires precisely coordinated changes in gene expression, but the underlying mechanisms remain unclear. The RNA-binding protein Trim71 is essential for development and serves as a potent regulator of post-transcriptional gene expression. Here, we show that global deficiency of *Trim71* induces severe defects in mesoderm-derived cells at the onset of organogenesis. Murine *Trim71*-KO embryos displayed impaired primitive erythropoiesis, yolk sac vasculature, heart function, and circulation, explaining the embryonic lethality of these mice. Tie2^Cre *Trim71* conditional knockout did not induce strong defects, showing that Trim71 expression in endothelial cells and their immediate progenitors is dispensable for embryonic survival. scRNA-seq of E7.5 global *Trim71*-KO embryos revealed that transcriptomic changes arise already at gastrulation, showing a strong up-regulation of the mesodermal pioneer transcription factor Eomes. We identify Eomes as a direct target of Trim71-mediated mRNA repression via the NHL domain, demonstrating a functional link between these important regulatory genes. Taken together, our data suggest that Trim71-dependent control of gene expression at gastrulation establishes a framework for proper development during organogenesis.**

## Introduction

The circulatory system consists of blood vessels, the heart, and blood cells, and it mediates the exchange of oxygen, nutrients, and cells throughout the body. The components of the circulatory system develop at the onset of organogenesis, starting at E8.5 in mice (Bautch & Caron, 2015). Cardiovascular development and the generation of blood cells from hematopoiesis are essential for mammalian embryonic survival during organogenesis (Dumont et al, 1994; Shalaby et al, 1995; Fujiwara et al, 1996; Koushik et al, 2001). Cells of the circulatory system are derived from the mesodermal germ layer, which emerges from cells passing through the primitive streak during gastrulation at E6.5–E7.5 in murine embryonic development (Bardot & Hadjantonakis, 2020; Prummel et al, 2020). The mesoderm gives rise to the hematoendothelial lineage via precursors harboring the potential to differentiate into endothelial cells (EC) and primitive erythroid cells (EryP) (Ema et al, 2006; Stefanska et al, 2017; Biben et al, 2023). Trajectory analysis of single-cell RNA sequencing (scRNA-seq) data from whole mouse embryos identified a cell population preceding EC and EryP differentiation, termed hematoendothelial progenitors (HEP), that is present from E7.0 to E8.25 (Pijuan-Sala et al, 2019). Within the extraembryonic yolk sac, the aggregation of EC leads to the de novo generation of a primary capillary plexus (vasculogenesis, E7.5–E8.5) (Chong et al, 2011), which is subsequently remodeled into a functional vascular network by various angiogenic processes including vessel sprouting, pruning, and increase in vessel diameter (angiogenesis, from E8.5 on) (Garcia & Larina, 2014). Cardiomyocytes and cardiac EC are also derived from the mesodermal germ layer (Srivastava & Olson, 2000), and blood circulation is gradually established after the onset of cardiac contractility around E8.0 (McGrath et al, 2003; Lucitti et al, 2007). Primitive erythropoiesis from precursor cells expressing endothelial surface markers at late gastrulation leads to the generation of EryP, which mediate oxygen transport and remain the only erythroid cell population until E11.5 in mice (Ema et al, 2006; Lux et al, 2008; McGrath & Palis, 2008; Pijuan-Sala et al, 2019; Iturri et al, 2021). EryP are first present in the yolk sac and disseminate through the embryo body in parallel to the establishment of blood circulation after the onset of heart function (McGrath et al, 2003). Transient definitive hematopoietic cells also emerge in the yolk sac by the differentiation of hemogenic endothelial cells into erythro-myeloid progenitors (EMP) at E8.5 (McGrath et al, 2015; Kasaai et al, 2017). EMP give rise to pre-macrophages (pMac) that

[1]Molecular Immunology and Cell Biology, Life & Medical Sciences Institute (LIMES), University of Bonn, Bonn, Germany [2]Quantitative Systems Biology, Life & Medical Sciences Institute (LIMES), University of Bonn, Bonn, Germany [3]Genomics and Immunoregulation, Life & Medical Sciences Institute (LIMES), University of Bonn, Bonn, Germany [4]Systems Medicine, Deutsches Zentrum für Neurodegenerative Erkrankungen (DZNE) e.V., Bonn, Germany [5]PRECISE Platform for Genomics and Epigenomics, Deutsches Zentrum für Neurodegenerative Erkrankungen (DZNE) e.V. and University of Bonn and West German Genome Center, Bonn, Germany [6]Immunogenomics and Neurodegeneration, Deutsches Zentrum für Neurodegenerative Erkrankungen (DZNE) e.V., Bonn, Germany [7]Developmental Biology of the Immune System, Life & Medical Sciences Institute (LIMES), University of Bonn, Bonn, Germany

Correspondence: wkolanus@uni-bonn.de

◢◣◤ Life Science Alliance

exit the yolk sac via the vasculature and translocate into the embryo proper, where they give rise to intraembryonic macrophage populations (Mass et al, 2016; Stremmel et al, 2018).

Recent studies have highlighted the importance of RNA-binding proteins in cardiovascular development (Völkers et al, 2024). The RNA-binding protein Trim71 is an essential and conserved regulator of embryonic development (Lin et al, 2007; Ecsedi et al, 2015; Mitschka et al, 2015). At the molecular level, Trim71 controls post-transcriptional gene expression by the interaction of its NHL domain with secondary structures in the 3′ untranslated region (3′ UTR) of mRNAs that have been termed Trim71-responsive elements (TREs) (Kumari et al, 2018; Torres-Fernández et al, 2019). Target mRNA binding by Trim71 leads to their degradation, thus repressing gene expression (Loedige et al, 2013). Global knockout of *Trim71* (*Trim71*-KO) in mice results in embryonic lethality at E9.5–E11.5 (Maller Schulman et al, 2008; Cuevas et al, 2015; Mitschka et al, 2015). *Trim71*-KO embryos display a cranial neural tube closure defect, but it has been argued before that this is presumably not the cause of lethality (Ecsedi & Grosshans, 2013). The underlying reason for the embryonic lethality upon *Trim71*-KO is so far not understood. Although the molecular functions of Trim71 have been extensively studied in vitro (Worringer et al, 2014; Welte et al, 2023), the role of Trim71 in mammalian embryonic development in vivo is largely unexplored. So far, a function of mammalian Trim71 has been described in neurogenesis and germ line development (Chen et al, 2012; Du et al, 2020; Torres-Fernández et al, 2021). Mutations in the human *TRIM71* gene cause congenital hydrocephalus, highlighting the relevance of this gene for human prenatal development (Duy et al, 2022, 2024). Considering the widespread expression of *Trim71* at gastrulation and early organogenesis (Chen et al, 2012), it is plausible that Trim71 also has functions in the development of other cell types, beyond neural and germ cells, that have not yet been described.

In the present study, we identify Trim71 as an essential factor for primitive erythropoiesis and cardiovascular development, explaining the embryonic lethality of murine *Trim71*-KO embryos. Surprisingly, the expression of *Trim71* in EC and their immediate *Tie2*-expressing progenitors is largely dispensable for cardiovascular development and EryP generation. Instead, we show that *Trim71*-KO results in extensive transcriptional changes in the mesoderm at E7.5 that directly precede the onset of defects in the hematoendothelial cell lineage. Mechanistically, Trim71 antagonizes the expression of the mesodermal pioneer transcription factor Eomes and binds to Eomes mRNA in an NHL domain–dependent manner, indicating Trim71-mediated post-transcriptional repression of *Eomes*. Our results delineate novel functions of Trim71 in the mesoderm and indicate that defects in the development of the circulatory system can be initiated at gastrulation.

# Results

## Impaired primitive erythropoiesis and vascular development in *Trim71*-KO embryos

To characterize the onset and progression of developmental phenotypes caused by global deficiency of Trim71, the morphology

of *Trim71*-KO embryos was evaluated by light microscopy at E7.5–E10.5. *Trim71*-KO embryos were morphologically indistinguishable from wild-type littermate control (WT) embryos at E7.5 and E8.5 (Fig 1A). In agreement with previous observations, *Trim71*-KO embryos were smaller than WT embryos at E9.5 and displayed a cranial neural tube closure defect (Fig 1A) (Mitschka et al, 2015). These phenotypes were even more pronounced at E10.5, at which stage *Trim71*-KO embryos showed severe underdevelopment across the whole body (Fig 1A). The growth retardations after E9.5 were underscored by decreased cell numbers in the yolk sac, embryo head, and embryo body (Fig S1A–C). In addition to these previously described phenotypes, we observed that *Trim71*-KO embryos appear pale in color from E9.5 on. This was in particular noticeable in the heart and the dorsal aorta, which are normally red in color in WT embryos because of the presence of EryP (Fig 1A). This prompted us to further analyze the yolk sac of *Trim71*-KO embryos, which is the origin of all erythroid cells before E11.5 (McGrath & Palis, 2008). Likewise, *Trim71*-KO yolk sacs also appeared pale (Fig 1B) and a quantification of relative Ter119$^+$ CD45$^-$ EryP numbers by flow cytometry at E9.5 and E10.5 revealed a strong reduction of EryP in *Trim71*-KO compared with WT yolk sacs (Fig 1C and D). A significant decrease in EryP numbers was also observed in the embryo head and embryo body at E10.5 (Fig 1E and F). The surface expression of the transferrin receptor CD71 on yolk sac EryP was unchanged between genotypes (Fig S1D and E), indicating that the limited amount of EryP that are produced in *Trim71*-KO embryos display normal maturation marker expression (Fraser et al, 2007). Altogether, these data show that *Trim71*-KO leads to a decrease in primitive erythropoiesis.

The yolk sac is an early site of vascular development and is an integral component of the embryonic circulatory system. After E9.5, the yolk sac vasculature is composed of large blood vessels, also known as vitelline vessels, and microvascular areas (Garcia & Larina, 2014). Closer inspection of the light microscopy images of E9.5 and E10.5 *Trim71*-KO yolk sacs revealed abnormal vascular structures (Fig 1B). We further investigated the yolk sac vasculature by whole-mount immunofluorescence staining using the EC marker CD31. Overview microscopy images showed that *Trim71*-KO yolk sacs were completely devoid of large vitelline blood vessels and had instead only equally sized small blood vessels (Fig 2A). Moreover, higher magnification imaging of the yolk sac microvasculature showed clear structural differences in the vascular network upon *Trim71*-KO (Fig 2B). We quantified endothelial extensions and vascular branching points in the microvasculature as markers of yolk sac angiogenesis. Endothelial extensions, defined as CD31$^+$ structures emerging from one vessel but not yet connected to another vessel (Fig 2B, magnification), were strongly reduced upon *Trim71*-KO (Fig 2C). These endothelial extensions arise from newly sprouting vessels or regressing vessels, representing two key angiogenic processes (Jones et al, 2008). Likewise, *Trim71*-KO led to a significant reduction in branching points, defined as the intersection of at least three vessels (Fig 2B, magnification, Fig 2D). Flow cytometric quantification of CD31$^+$ AA4.1$^-$ yolk sac EC and CD31$^+$ EC in the embryo head and embryo body showed, however, no differences in EC numbers upon *Trim71*-KO (Fig S1F–H). Taken together, these data demonstrate that *Trim71*-KO embryos have a yolk sac

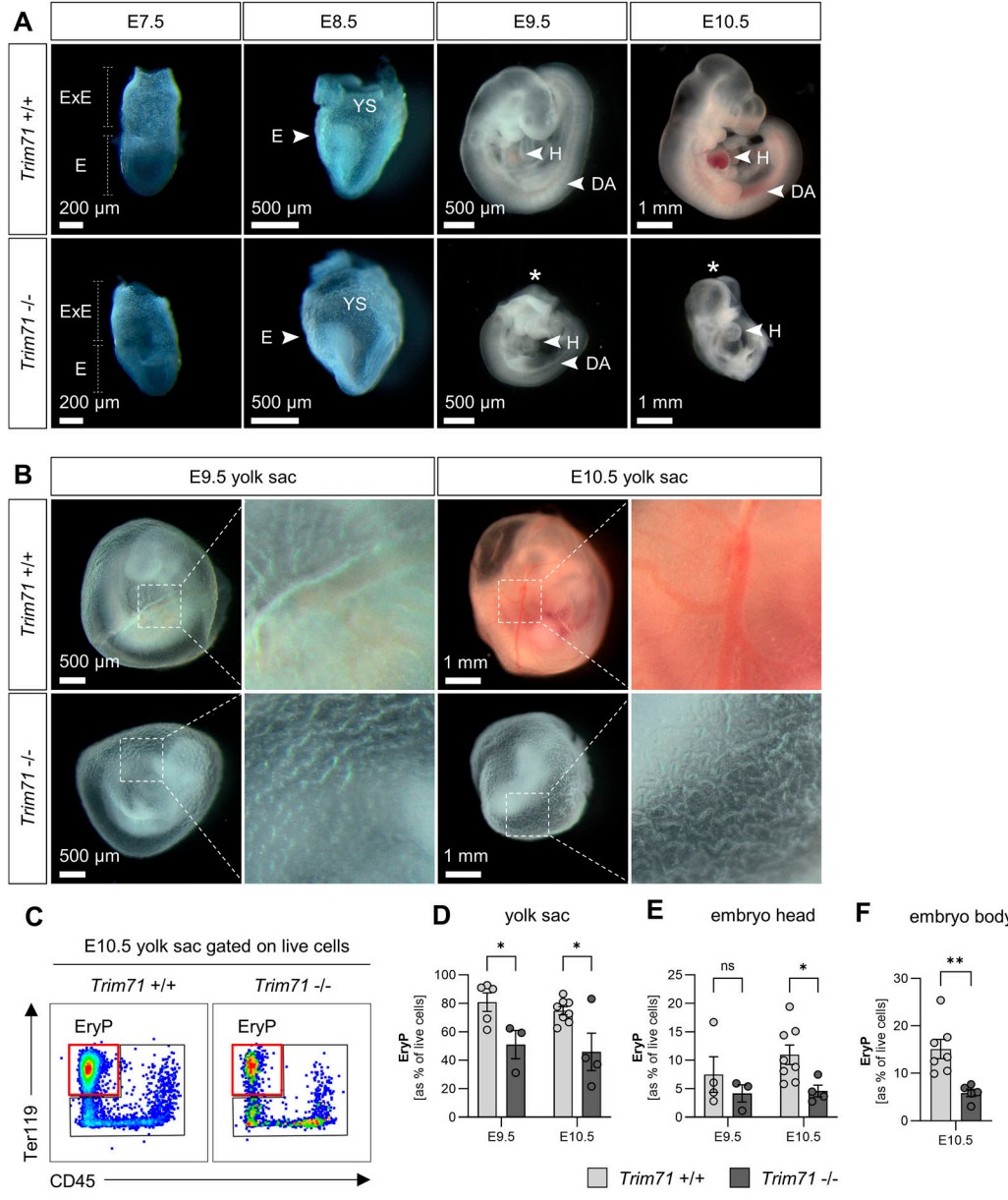

**Figure 1. *Trim71*-KO embryos show developmental retardation and decreased primitive erythropoiesis at early organogenesis.**
**(A)** Morphology of WT and *Trim71*-KO embryos at indicated developmental stages E7.5–E10.5. E, embryo; ExE, extraembryonic region; DA, dorsal aorta; H, heart; YS, yolk sac. Stars indicate the presence of a neural tube closure defect. **(B)** Morphology of WT and *Trim71*-KO yolk sacs at E9.5 and E10.5. Dashed boxes show the magnification of vitelline vessels. **(C)** Representative flow cytometry gating of Ter119$^+$ CD45$^-$ EryP in E10.5 WT and *Trim71*-KO yolk sacs. Red boxes indicate gates for EryP. **(D, E, F)** Relative quantification of EryP in the (D) yolk sac and (E) embryo head at E9.5 and E10.5, and (F) the embryo body at E10.5 by flow cytometry (n = 3–8 embryos from 2 to 3 experiments; data are depicted as the mean ± SEM, unpaired *t* test, *$P < 0.05$, **$P < 0.01$).

vascular remodeling defect, whereas vasculogenesis per se appears to be intact. We further investigated whether the intra-embryonic circulatory system is affected by *Trim71*-KO. *Trim71*-KO embryos form a heart (Fig 1A), but 50% of E9.5 and 20% of E10.5 *Trim71*-KO embryos had no visible heartbeat (Fig 2E, Video 1 and Video 2). Furthermore, the subset of E10.5 *Trim71*-KO embryos that had a heartbeat showed a significant reduction in heart rate compared with WT embryos (Fig 2F, Video 3). In summary, *Trim71*-KO embryos display defects in all major components of the circulatory system.

## Defects in the circulatory system result in impaired vascular pMac migration in *Trim71*-KO embryos

In order to investigate the functional relevance of the circulatory system defects in *Trim71*-KO embryos, we analyzed the migration efficiency of EMP-derived pMac from the yolk sac to the embryo proper, where they give rise to intraembryonic macrophage populations (Mass et al, 2016). The migration of pMac occurs through the yolk sac vasculature and depends on proper heart function (Ginhoux et al, 2010; Stremmel et al, 2018). We quantified transient

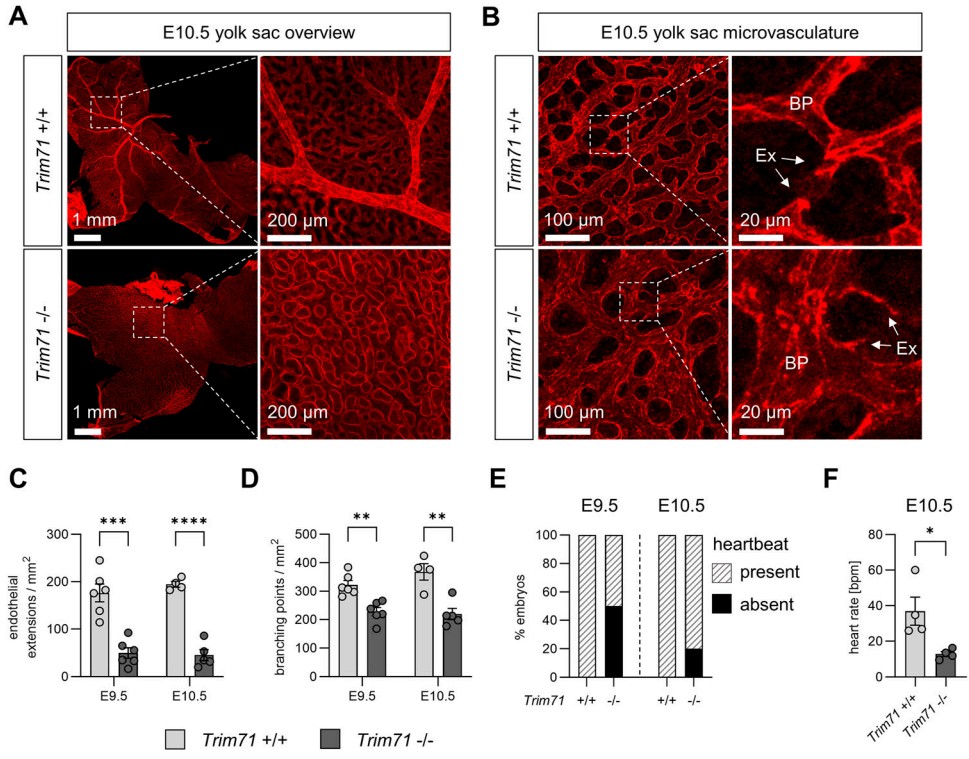

**Figure 2. Cardiovascular defects in *Trim71*-KO embryos.**
**(A)** Representative overview images of E10.5 yolk sacs stained with the EC marker CD31. Dashed boxes show the magnification of vitelline vessels in the *Trim71*-WT yolk sac, whereas a representative region devoid of vitelline vessels is shown in the *Trim71*-KO yolk sac. **(B)** Representative images of the yolk sac microvasculature at E10.5 stained with CD31. Dashed boxes show the magnification of individual vessels with indicated endothelial extensions and branching points. BP, branching point; Ex, endothelial extension. **(C, D)** Quantification of (C) endothelial extensions and (D) branching points in the yolk sac microvasculature at E9.5 and E10.5 (n = 4–6 yolk sacs from three experiments per stage; data are depicted as the mean ± SEM, unpaired *t* test, **P < 0.01, ***P < 0.001, ****P < 0.0001). **(E)** Percentage of embryos with present or absent heartbeat at E9.5 and E10.5 (n = 8–25 embryos from 5 to 14 experiments). **(F)** Heart rate of E10.5 WT and *Trim71*-KO embryos in which a heartbeat was detectable (n = 4 embryos from 3 experiments; data are depicted as the mean ± SEM, unpaired *t* test, *P < 0.05).

definitive hematopoietic cells and their derivatives in the yolk sac, embryo body, and embryo head by flow cytometry. In *Trim71*-KO yolk sacs, EMP numbers (CD45$^{low}$ Kit$^+$ AA4.1$^+$) were slightly increased at E9.5 and pMac numbers (CD45$^+$ Kit$^-$ CD11b$^+$ F4/80$^-$) were elevated at E10.5, whereas yolk sac macrophage numbers (CD45$^+$ Kit$^-$ CD11b$^+$ F4/80$^+$) were unaffected (Fig 3A). In contrast, pMac and macrophages were markedly reduced in the body and were almost completely absent from the head of *Trim71*-KO embryos (Fig 3B and C). These data indicate that transient definitive hematopoietic cells emerge normally in *Trim71*-KO yolk sacs, but pMac fail to migrate from the yolk sac to the embryo proper. Besides cardiovascular defects, impaired macrophage progenitor migration can also result from hematopoietic cell–intrinsic defects, for example, caused by the loss of the chemokine receptor Cx3cr1 (Mass et al, 2016). To address this possibility, we first analyzed *Trim71* expression in transient definitive hematopoietic cells of WT embryos. Indeed, we found that *Trim71* is expressed by EMP and pMac, whereas its expression declines upon differentiation into macrophages (Fig 3D). We therefore tested whether targeted deletion of *Trim71* in EMP and their progeny, using the *Csf1r*$^{iCre}$ driver line and the *Trim71*-flox line, has an effect on EMP-derived hematopoiesis or intraembryonic macrophage colonization. *Csf1r*$^{iCre}$ *Trim71* conditional knockout (cKO) embryos did not show any differences in the number of EMP, pMac, and macrophages in the yolk sac and embryo head at E9.5 and E10.5 compared with control embryos (Fig 3E and F). Moreover, *Csf1r*$^{iCre}$ *Trim71* cKO pMac and macrophages showed no changes in the expression of Cx3cr1 (Fig S2A–C). *Csf1r*$^{iCre}$ *Trim71* cKO embryos also displayed no obvious morphological differences and were not lethal (Fig S2D and E). These data demonstrate that the expression

of *Trim71* in EMP is not required for intraembryonic macrophage colonization, and rule out an EMP- or pMac-intrinsic origin of the impaired vascular migration of pMac from the yolk sac into the embryo proper upon global *Trim71*-KO.

## Yolk sac scRNA-seq identifies transcriptional changes in *Trim71*-KO EC

To analyze changes in gene expression associated with impaired vascular development of *Trim71*-KO embryos, we performed single-cell mRNA sequencing (scRNA-seq) of whole E9.5 WT and *Trim71*-KO yolk sacs. We identified the cell types expected to be present in the yolk sac, including EC and EryP (Figs 4A and S3A). Fitting to the flow cytometry data (Fig 1D), EryP were proportionally decreased upon *Trim71*-KO in the scRNA-seq dataset (Fig S3B and C). Uniform Manifold Approximation and Projection (UMAP) showed differential localization of cells from the same cell type between genotypes, indicating widespread transcriptomic changes in *Trim71*-KO yolk sacs (Fig 4B). Accordingly, analysis of differentially expressed genes (DEG) between genotypes identified extensive changes in gene expression upon *Trim71*-KO in all cell types, including EC, EryP, and the extraembryonic endoderm (ExE endoderm) (Fig S3D–F). Gene ontology (GO) analysis of the 430 down-regulated DEG in EC revealed an enrichment of genes in multiple processes related to angiogenesis (Fig 4C). Down-regulated genes contained in the process *regulation of angiogenesis* included the endothelial transcription factor Ets1, which is known to play a role in vascular development (Wei et al, 2009), and the blood flow–induced transcription factor Klf2 (Lee et al, 2006) (Fig 4D). Moreover, *Trim71*-KO EC had decreased expression of the cell junction protein Cdh5 (Fig 4D),

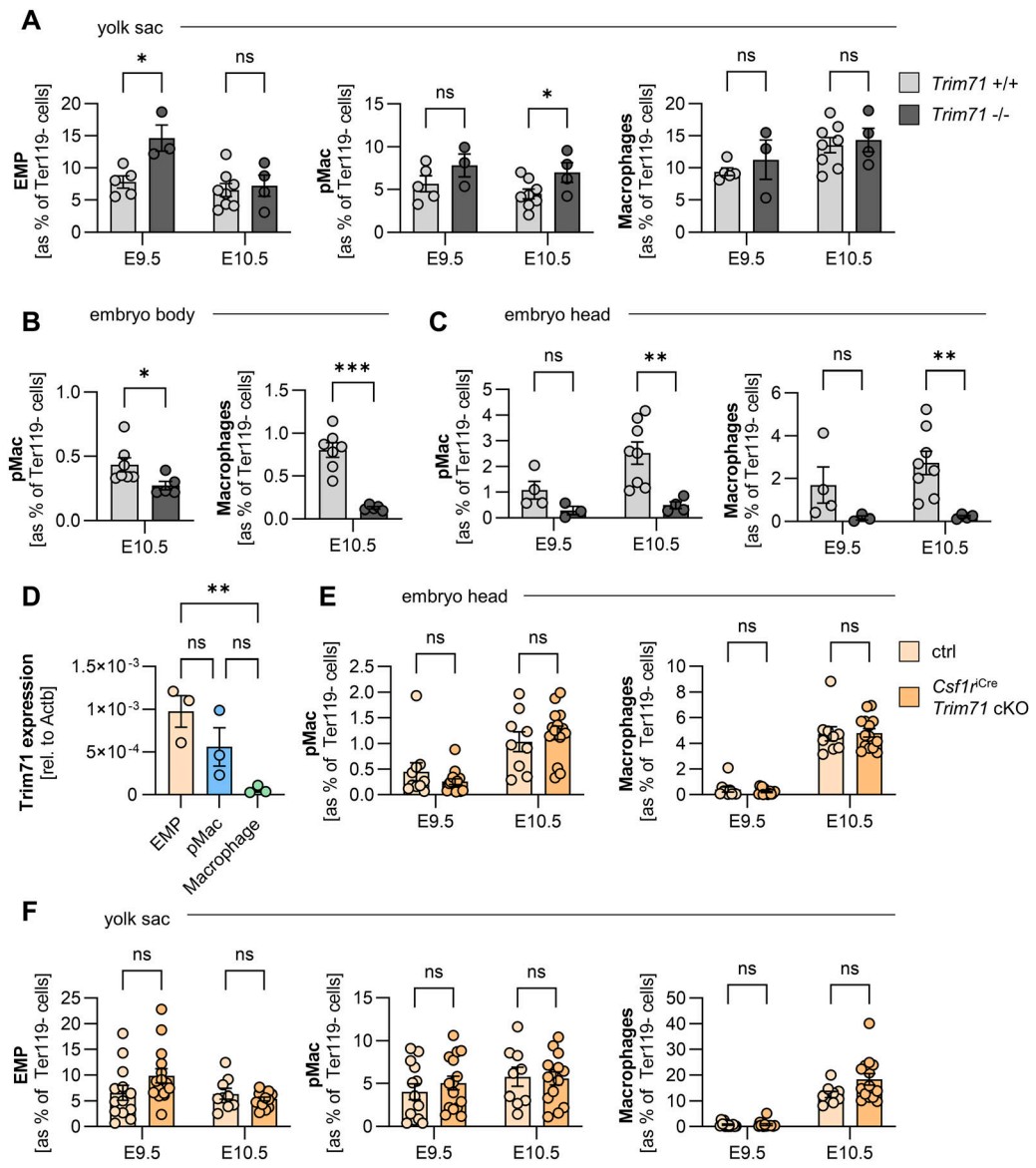

**Figure 3. Effect of global or erythro-myeloid progenitor (EMP)–specific *Trim71* deletion on EMP-derived myeloid cell numbers in the yolk sac and the embryo.**
**(A, B, C)** Quantification of cell numbers in WT and *Trim71*-KO embryos. **(A)** Relative numbers of EMP, pMac, and macrophages in the yolk sac at E9.5 and E10.5.
**(B, C)** Relative numbers of pMac and macrophages in the embryo body at E10.5 and (C) in the embryo head at E9.5 and E10.5 (n = 3–8 embryos from 2 to 3 experiments; data are depicted as the mean ± SEM, unpaired *t* test, ns, not significant, \*P < 0.05, \*\*P < 0.01). **(D)** Quantification of Trim71 mRNA expression by qRT–PCR in EMP, pMac, and macrophages isolated from WT E10.5 yolk sacs (n = 3 from 3 experiments; data are depicted as the mean ± SEM, ordinary one-way ANOVA, ns, not significant, \*\*P < 0.01).
**(E, F)** Quantification of cell numbers in control and *Csf1r*iCre *Trim71* cKO (*Csf1r*iCre/+ *Trim71*fl/fl) embryos. Ctrl indicates *Csf1r*+/+ *Trim71*fl/fl. **(E)** Relative numbers of pMac and macrophages in the embryo head at E9.5 and E10.5. **(F)** Relative numbers of EMP, pMac, and macrophages in the yolk sac at E9.5 and E10.5 (n = 9–15 embryos from 4 to 5 experiments; data are depicted as the mean ± SEM, unpaired *t* test, ns, not significant).

which is required for proper angiogenesis and can be transactivated by Ets1 (Lelièvre et al, 2000; Bentley et al, 2014). In line with the decreased endothelial extensions in *Trim71*-KO yolk sacs (Fig 2C), the gene expression score of genes involved in s*prouting angiogenesis* was significantly decreased in *Trim71*-KO EC (Fig 4E). Visualization of GO processes from down-regulated genes in EC by a category network plot showed the presence of four clusters, which were related to cell migration, endothelial cell differentiation, cell junction assembly, and regulation of vascular development (Fig 4F). These data indicate that the

impaired yolk sac vascular remodeling of *Trim71*-KO embryos is the result of multiple distinct EC-intrinsic processes involved in angiogenesis.

## Trim71 expression in the mesoderm and in the hematoendothelial lineage

We next sought to investigate the developmental origins leading to the impaired vascular development and defective primitive erythropoiesis of *Trim71*-KO embryos. To this end, we examined

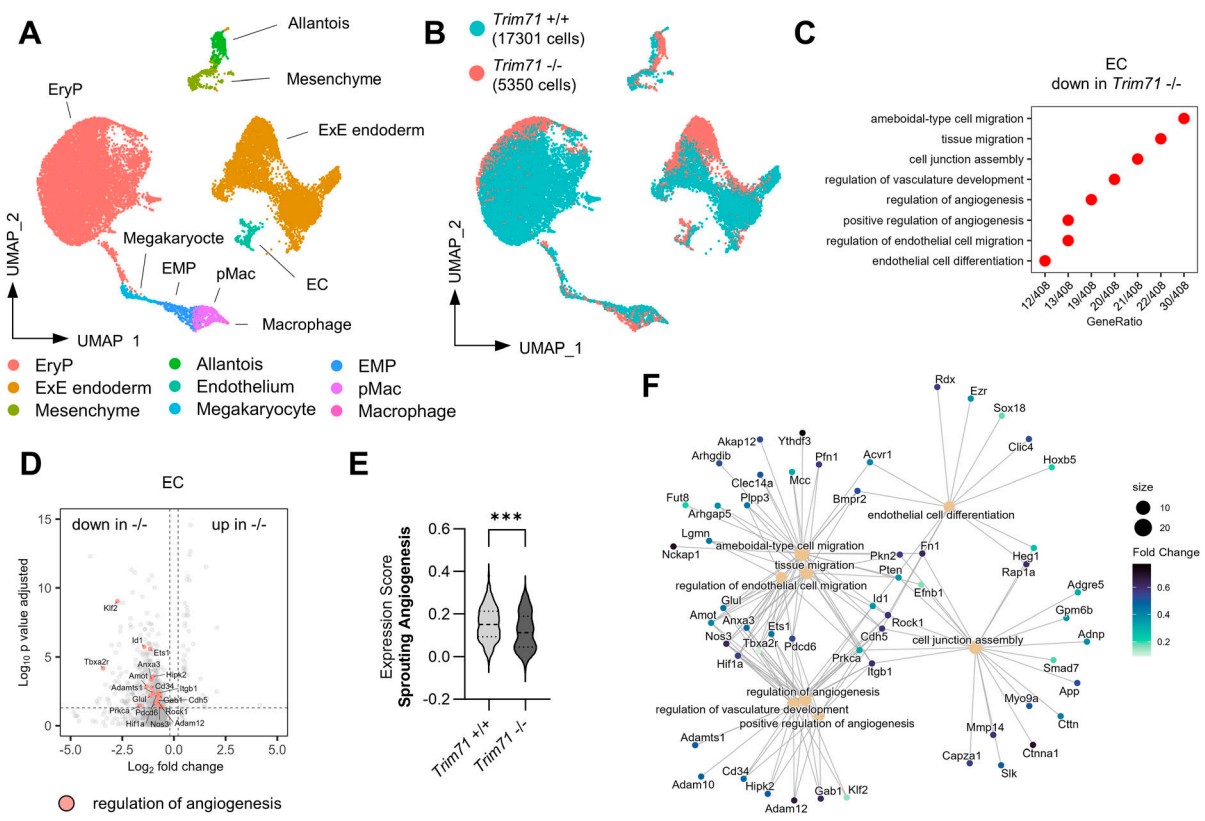

**Figure 4. ScRNA-seq of E9.5 yolk sacs reveals the decreased endothelial expression of angiogenic genes upon *Trim71*-KO.**
**(A)** Uniform Manifold Approximation and Projection plot of all cells with color-coded cell-type annotations. **(B)** Uniform Manifold Approximation and Projection plot with color-coded genotypes (blue = *Trim71* +/+, red = *Trim71* −/−). **(C)** Enriched GO terms among down-regulated differentially expressed genes (DEG) in *Trim71* −/− EC. **(D)** Volcano plot of DEG in *Trim71* −/− EC with genes included in the GO-term *regulation of angiogenesis* highlighted in red. **(E)** Expression score of the MSigDB gene set *sprouting angiogenesis* (data are depicted as a violin plot with median and quartiles indicated as dashed lines, unpaired *t* test, ***P < 0.001). **(F)** Category network plot of GO terms from DEG down-regulated in *Trim71* −/− EC with color-coded expression fold changes of genes included in the processes. The circle size denotes the number of DEG contained in each process.

Trim71 expression from gastrulation to early organogenesis with a focus on the hematoendothelial cell lineage. Analysis of scRNA-seq data from E6.5–E8.5 WT mouse embryos (Pijuan-Sala et al, 2019) by pseudo-bulk expression analysis across cell types showed that *Trim71* is highly expressed in the primitive streak and nascent mesoderm at E7.5 (Fig S4A). Intermediate *Trim71* expression levels were present in HEP and EC, whereas *Trim71* expression was absent in EryP (Fig S4A). Using immunofluorescence staining, we found the ubiquitous embryonic protein expression of Trim71 at E7.5, including Trim71 expression in Flk1⁺ cells that encompass both mesodermal progenitors and HEP (Biben et al, 2023) (Fig S4B). Moreover, Trim71 protein expression was also present in CD31⁺ embryonic EC of the yolk sac at E9.5 and the dorsal aorta at E10.5 (Fig S4C and D). Of note, although the Trim71 protein signal was diffuse cytoplasmic in E7.5 embryos and CD31⁺ cells of the dorsal aorta, Trim71 protein expression in the yolk sac was restricted to discrete foci within CD31⁺ EC and also in CD31⁻ cells, reminiscent of previously observed Trim71 dot-like staining patterns in cell lines (Torres-Fernández et al, 2021).

### *Tie2*^Cre-induced *Trim71* deletion does not phenocopy *Trim71*-KO

Because we detected the expression of Trim71 in EC and HEP, we investigated the function of Trim71 in the hematoendothelial lineage by conditional knockout induced via the *Tie2*^Cre driver line (*Tie2*^Cre *Trim71* cKO). Tie2 is expressed by differentiated EC and their precursors that have the potential to give rise to both endothelial and erythroid cells (Ema et al, 2006; Stefanska et al, 2017), and consequently, the *Tie2*^Cre line targets both EC and EryP (Kisanuki et al, 2001; Tang et al, 2010). At E10.5, *Tie2*^Cre *Trim71* cKO embryos and yolk sacs appeared normal in overall morphology (Fig 5A). Mendelian ratios of *Tie2*^Cre *Trim71*^fl/fl were retrieved at developmental stages E9.5–E12.5, and viable *Tie2*^Cre *Trim71*^fl/fl mice were born, showing that *Tie2*^Cre *Trim71* cKO does not lead to embryonic lethality (Fig 5B). In the yolk sac, *Tie2*^Cre *Trim71* cKO did not prevent the formation of large vitelline vessels (Fig 5A and C). We detected a slight decrease in endothelial extensions and branching points in the yolk sac microvasculature upon *Tie2*^Cre *Trim71* cKO at E9.5 (Fig 5D and E); these effects were, however, mild compared with global *Trim71*-KO (Fig 2B–D). At E12.5, endothelial extensions remained significantly reduced in *Tie2*^Cre *Trim71* cKO yolk sacs, whereas

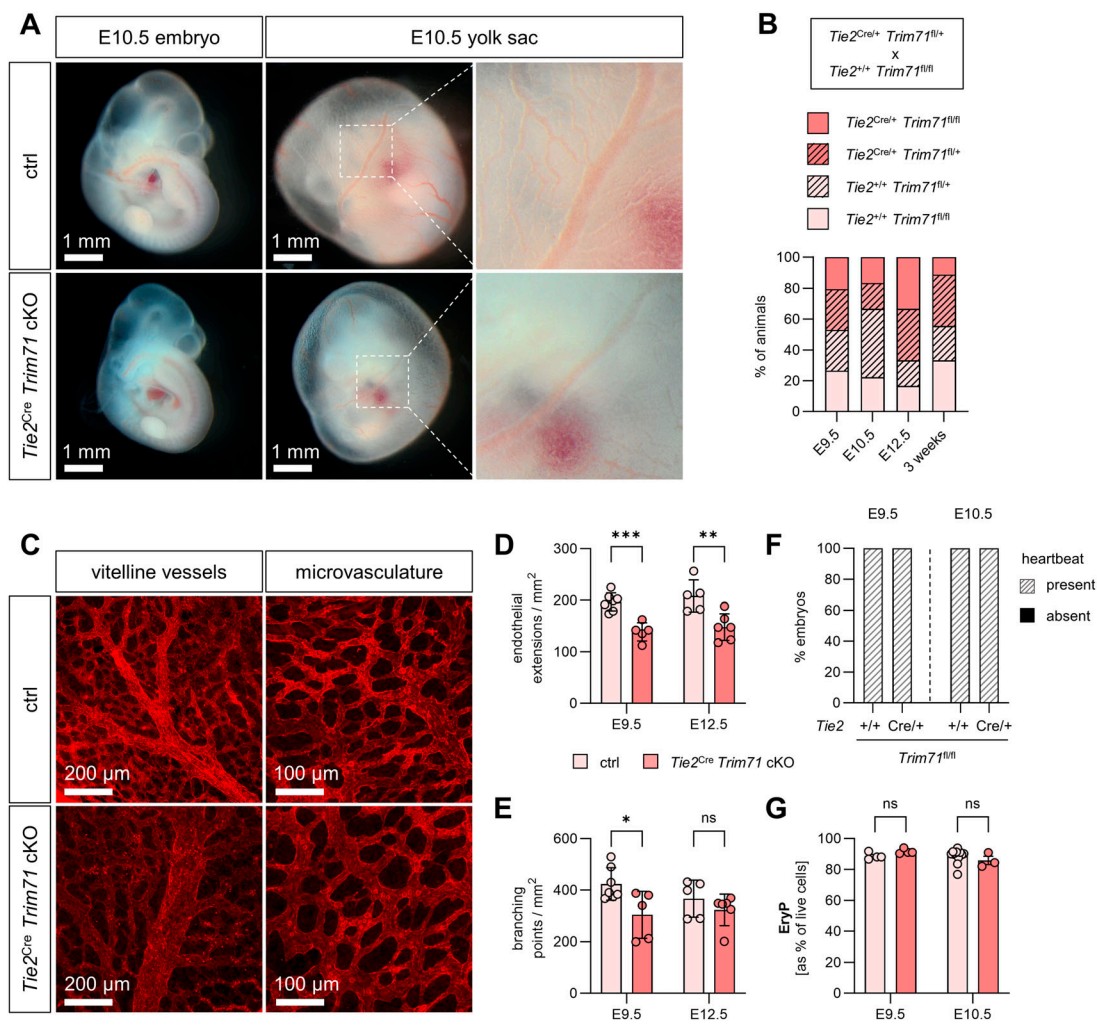

**Figure 5. Tie2<sup>Cre</sup> Trim71 cKO does not result in embryonic lethality and does not induce strong defects in cardiovascular development or primitive erythropoiesis.**
(A) Morphology of control or *Tie2*<sup>Cre</sup> *Trim71* cKO (*Tie2*<sup>Cre/+</sup> *Trim71*<sup>fl/fl</sup>) embryos and yolk sacs at E10.5. Ctrl indicates *Tie2*<sup>+/+</sup> *Trim71*<sup>fl/fl</sup> or *Tie2*<sup>+/+</sup> *Trim71*<sup>fl/+</sup>. Dashed boxes show the magnification of vitelline vessels. (B) Quantification of genotype percentages in E9.5, E10.5, and E12.5 embryos and 3-wk-old pups from mating of *Tie2*<sup>Cre/+</sup> *Trim71*<sup>fl/+</sup> with *Tie2*<sup>+/+</sup> *Trim71*<sup>fl/fl</sup> mice (n = 9–34 embryos or pups from 2 to 5 litters. (C, D, E) Whole-mount staining of yolk sacs with CD31. (C) Representative images of vitelline vessels and microvascular areas of E9.5 yolk sacs. (D, E) Quantification of (D) endothelial extensions and (E) branching points in the microvasculature of E9.5 and E12.5 yolk sacs (n = 5–6 yolk sacs from 3 to 5 experiments; data are depicted as the mean ± SEM, unpaired t test, ns, not significant, *P < 0.05, **P < 0.01, ***P < 0.001). (F) Percentage of embryos with present or absent heartbeat at E9.5 and E10.5 (n = 3–12 embryos from 2 to 4 experiments). (G) Relative quantification of EryP in the yolk sac at E9.5 and E10.5 by flow cytometry (n = 3–11 embryos from 1 to 2 experiments; data are depicted as the mean ± SEM, unpaired t test, ns, not significant).

branching points normalized to control embryo levels (Fig 5D and E). All *Tie2*<sup>Cre</sup> *Trim71* cKO embryos had a heartbeat at E9.5 and E10.5 (Fig 5F) and showed no changes in relative EryP numbers in the yolk sac (Fig 5G). These results demonstrate that the Trim71 expression in EC and *Tie2*-expressing precursors of EC and EryP is dispensable for primitive erythropoiesis and heart function and is only marginally required for yolk sac angiogenesis.

## Transcriptomic alterations in *Trim71*-KO embryos arise at gastrulation

The absence of strong vascular and erythropoiesis phenotypes in *Tie2*<sup>Cre</sup> *Trim71* cKO embryos led us to the hypothesis that the origin of the defects observed in *Trim71*-KO embryos could lie

earlier in embryonic development during gastrulation. We thus performed scRNA-seq of whole E7.5 *Trim71*-KO embryos. Cell-type annotation led to the identification of the expected gastrulation stage cell types, including cells from all three germ layers (Figs 6A and S5A). Analysis of *Trim71* +/+ cells showed high and ubiquitous expression of Trim71 across all germ layers (Fig S5D), in agreement with the ubiquitous Trim71 expression as observed by IF (Fig S4B). *Trim71*-KO embryos had slightly reduced numbers of ectodermal and definitive endodermal cell populations, whereas mesoderm cell numbers were equal between genotypes (Fig S5B and C). This demonstrates that the generation of mesodermal cells from gastrulation is not impaired by loss of *Trim71*. Interestingly, the ExE endoderm was markedly expanded in E7.5 *Trim71*-KO embryos and had an altered positioning in the

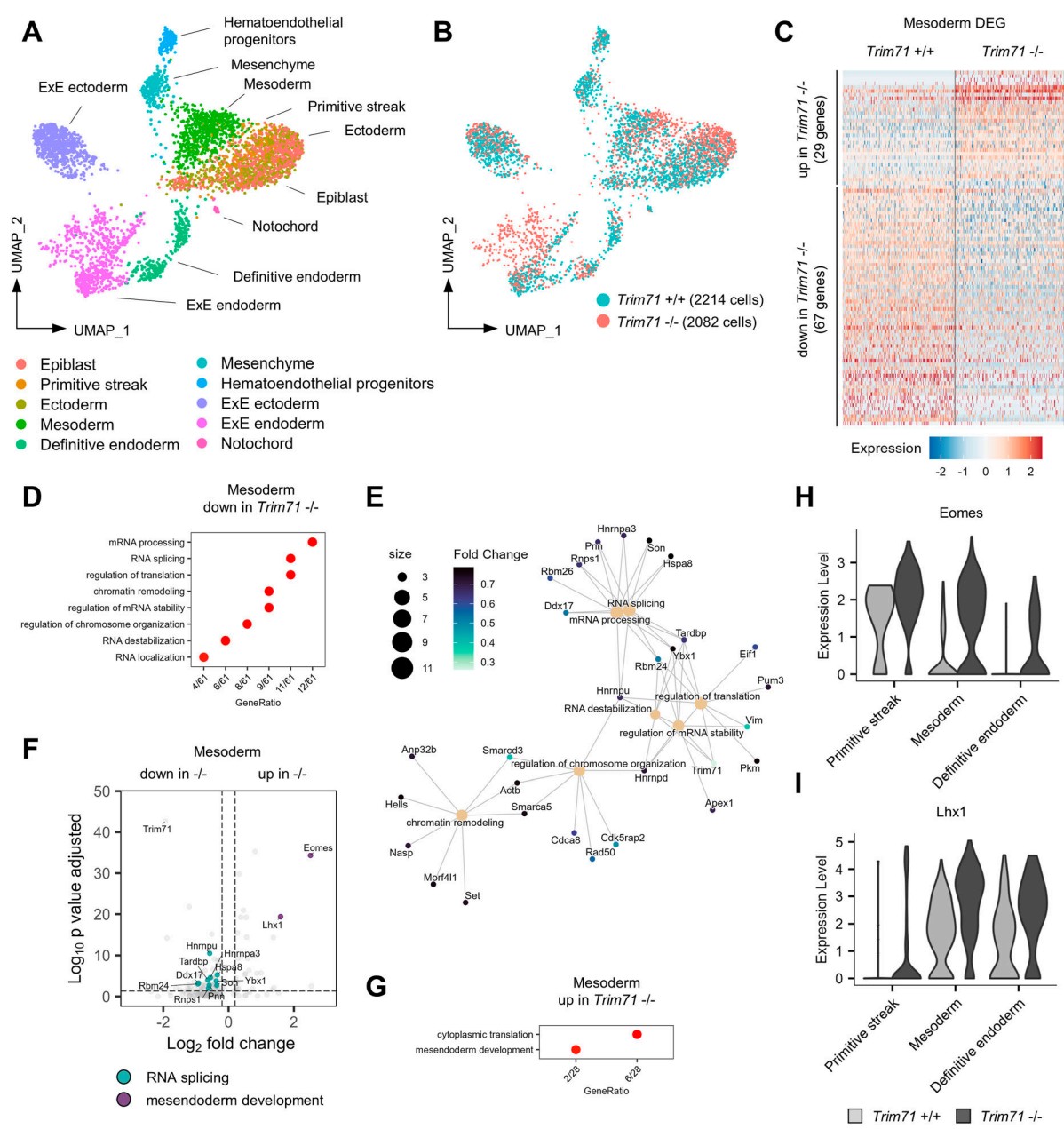

**Figure 6. scRNA-seq of E7.5 embryos reveals extensive transcriptional changes in the mesoderm upon *Trim71*-KO.**
**(A)** Uniform Manifold Approximation and Projection plot of all cells with color-coded cell-type annotations. **(B)** Uniform Manifold Approximation and Projection plot with color-coded genotypes (blue = *Trim71* +/+, red = *Trim71* −/−). **(C)** Expression heatmap of up-regulated and down-regulated differentially expressed genes (DEG) in *Trim71*+/+ and *Trim71*−/− mesodermal cells. **(D)** Enriched GO terms among down-regulated DEG in *Trim71*−/− mesodermal cells. **(E)** Category network plot of GO terms from DEG down-regulated in *Trim71*−/− mesodermal cells with color-coded expression fold changes of genes included in the processes. The circle size denotes the number of DEG contained in each process. **(F)** Volcano plot of DEG in *Trim71*−/− mesodermal cells with DEG included in the GO-term *RNA splicing* highlighted in turquoise and *mesendoderm development* in purple. **(G)** Enriched GO terms among up-regulated DEG in *Trim71*−/− mesodermal cells. **(H, I)** Expression of (H) Eomes and (I) Lhx1 in cells of the primitive streak, mesoderm, and definitive endoderm separated by genotype.

UMAP space compared with *Trim71*-WT cells, alongside extensive changes in gene expression (Figs 6B and S5C and E). We next focused on the mesoderm for differential gene expression analysis, because the hematoendothelial cell lineage is derived from this germ layer. *Trim71*-KO E7.5 mesodermal cells displayed transcriptomic changes, as evident by a slightly different relative position of WT and *Trim71*-KO mesodermal cells on the UMAP plot

(Fig 6B) and a substantial amount of DEG (Fig 6C, 29 up-regulated DEG, 67 down-regulated DEG). GO analysis of down-regulated genes in the *Trim71*-KO mesoderm yielded GO terms related to mRNA processing, RNA splicing, RNA stability, and chromatin remodeling (Fig 6D–F). Furthermore, we analyzed up-regulated genes in the *Trim71*-KO mesoderm to identify mRNAs that might be repressed by Trim71-mediated mRNA binding at gastrulation.

*Trim71*-KO resulted in the elevated mesodermal expression of the transcription factors Eomes and Lhx1, both of which play important roles in mesodermal development and were contained in the GO-term *mesendoderm development* (Fig 6F–I). The increased expression of Eomes was not only restricted to mesodermal cells, but was also present in cells of the primitive streak and definitive endodermal cells (Fig 6H). Lhx1 was not expressed in primitive streak cells of either genotype, but was increased upon *Trim71*-KO in mesodermal and definitive endodermal cells (Fig 6I). These data show that changes in mesodermal gene expression at late gastrulation precede the onset of defects in the circulatory system of *Trim71*-KO embryos.

### Trim71 antagonizes Eomes expression and binds to Eomes mRNA dependent on the NHL domain

We used *Trim71*-flox mESC, which express WT levels of Trim71, and *Trim71*-KO mESC as an in vitro model to study a potential direct regulation of Eomes and Lhx1 expression by Trim71. Cells were differentiated as embryoid bodies in the presence of serum for 4 d by the removal of leukemia inhibitory factor (LIF) and 2i (CHIR99021 and PD0325901) from the medium (Pearson et al, 2008). Flk1 mRNA expression was induced at day 4 of differentiation in *Trim71*-flox but not in *Trim71*-KO mESC, whereas Pdgfra mRNA expression was unaffected (Fig S6A and B). Analogous to the increased Eomes expression in *Trim71*-KO embryos at E7.5, Eomes mRNA levels were strongly elevated in *Trim71*-KO compared with *Trim71*-flox mESC at day 4 of differentiation (Fig 7A). In contrast, Lhx1 expression was unaltered between genotypes (Fig 7B). RNA secondary structure prediction showed the presence of a putative TRE within the 3′ UTR of the murine Eomes mRNA, which fulfills the structural requirements for interaction with the Trim71 NHL domain (Fig 7C) (Kumari et al, 2018; Torres-Fernández et al, 2019; Shi et al, 2024). The sequence of this TRE was UAU-CUUGGAGAUA, located at positions 3,452–3,465 within the Eomes mRNA (Fig 7D). In agreement with reported TRE characteristics (Kumari et al, 2018), the Eomes mRNA TRE consisted of a 13-mer stem–loop with a U-A base pair at the top of the stem and a G in position III of the loop (Fig 7D). To test direct binding of Eomes mRNA by Trim71, we performed cross-linking immunoprecipitation (CLIP) using mESC lines endowed with the endogenous expression of mNeon-FLAG-tagged *Trim71*-WT, *Trim71*-KO, or *Trim71*-R595H variants (Duy et al, 2022) at day 4 of differentiation (Fig 7E and F). The *Trim71*-R595H mutation is located within the NHL domain and abrogates the RNA-binding capacity of Trim71 (Duy et al, 2022). Flow cytometric analysis of Trim71 expression via the mNeon tag validated the absence of Trim71 expression in *Trim71*-KO mESC, whereas expression was retained in *Trim71*-WT and *Trim71*-R595H cells (Fig 7G and H). At day 4 of differentiation, *Trim71*-R595H cells showed a tendential up-regulation of Eomes mRNA levels (Fig S6C). Strikingly, FLAG-CLIP of Trim71 variants in differentiated mESC showed an enrichment of Eomes mRNA in *Trim71*-WT mESC, which was strongly reduced upon *Trim71*-KO and *Trim71*-R595H mutation (Fig 7I). Lhx1 mRNA was not reliably detected by qRT-PCR after CLIP (Fig 7I). Importantly, we validated Trim71 protein enrichment after immunoprecipitation (Fig 7J). Together, these data show that Trim71 selectively interacts with Eomes mRNA in an NHL domain–dependent manner, suggesting a regulation of this key mesodermal transcription factor by

the previously described mechanisms of Trim71-induced mRNA degradation (Loedige et al, 2013; Torres-Fernández et al, 2019).

## Discussion

### Trim71 is required for primitive erythropoiesis and cardiovascular development

In this study, we identify Trim71 as a crucial factor for the embryonic development and function of all major components of the circulatory system, including EryP, the vasculature, and the heart. In global *Trim71*-KO embryos, defects in the circulatory system become apparent with the onset of organogenesis at E9.5. We show that *Trim71*-KO embryos are noticeably pale and have strongly reduced levels of EryP in the yolk sac and in the embryo proper. Severe defects in vascular development were present in the yolk sac, as evident by the complete absence of large vitelline vessels and an unremodeled vascular network. Vasculogenesis was not noticeably impeded by loss of Trim71, but angiogenesis was severely decreased in *Trim71*-KO yolk sacs, as shown by decreased vascular branching points and a loss of endothelial extensions in the yolk sac microvasculature. These phenotypes are comparable to the yolk sac vascular remodeling defects of embryos deficient for crucial angiogenic factors, such as *Tie2*$^{-/-}$, *Nrp1*$^{-/-}$, or *Notch1*$^{-/-}$ embryos (Krebs et al, 2000; Tachibana et al, 2005; Jones et al, 2008). At the transcriptional level, scRNA-seq of E9.5 yolk sacs revealed decreased endothelial expression of genes involved in cell migration, cell junction assembly, and endothelial cell differentiation upon *Trim71*-KO. These processes are collectively required for vascular development, angiogenesis, and sprouting (Bazzoni & Dejana, 2004; Lamalice et al, 2007), providing an explanation for the loss of endothelial extensions and the yolk sac remodeling defect. Besides vascular impairments, *Trim71*-KO embryos display defects in heart function, as evident from a partial absence of heartbeat and a decreased heart rate. This was associated with defective colonization of the embryo proper by macrophages and their progenitors (pMac), which originate from the yolk sac. The presence of normal pMac and macrophage numbers in the embryo head of *Csf1r*$^{iCre}$ *Trim71* cKO embryos demonstrates that the migration defect of these cells in *Trim71*-KO embryos is independent of the Trim71 expression in EMP and pMac. This strongly indicates that the impaired heart function of these embryos leads to insufficient blood circulation to mediate the migration of pMac through the vasculature (Ginhoux et al, 2010; Stremmel et al, 2018), given that the embryonic heart rate correlates with blood flow velocity (Phoon et al, 2000). Because the vitelline vessels are the direct vascular connection between the yolk sac and the embryo, the atrophy of these blood vessels in *Trim71*-KO embryos might further enhance the retention of pMac within the yolk sac. An impaired blood circulation of *Trim71*-KO embryos and the reduced hematocrit because of the absence of EryP could in addition restrict vascular remodeling, which is known to be dependent on the hemodynamic forces of the blood (Koushik et al, 2001; Jones et al, 2008; Garcia & Larina, 2014). Accordingly, we observed the decreased expression of the blood flow–induced transcription factor Klf2 in *Trim71*-KO EC (Lee et al, 2006). Thus, impaired vascular remodeling in *Trim71*-KO

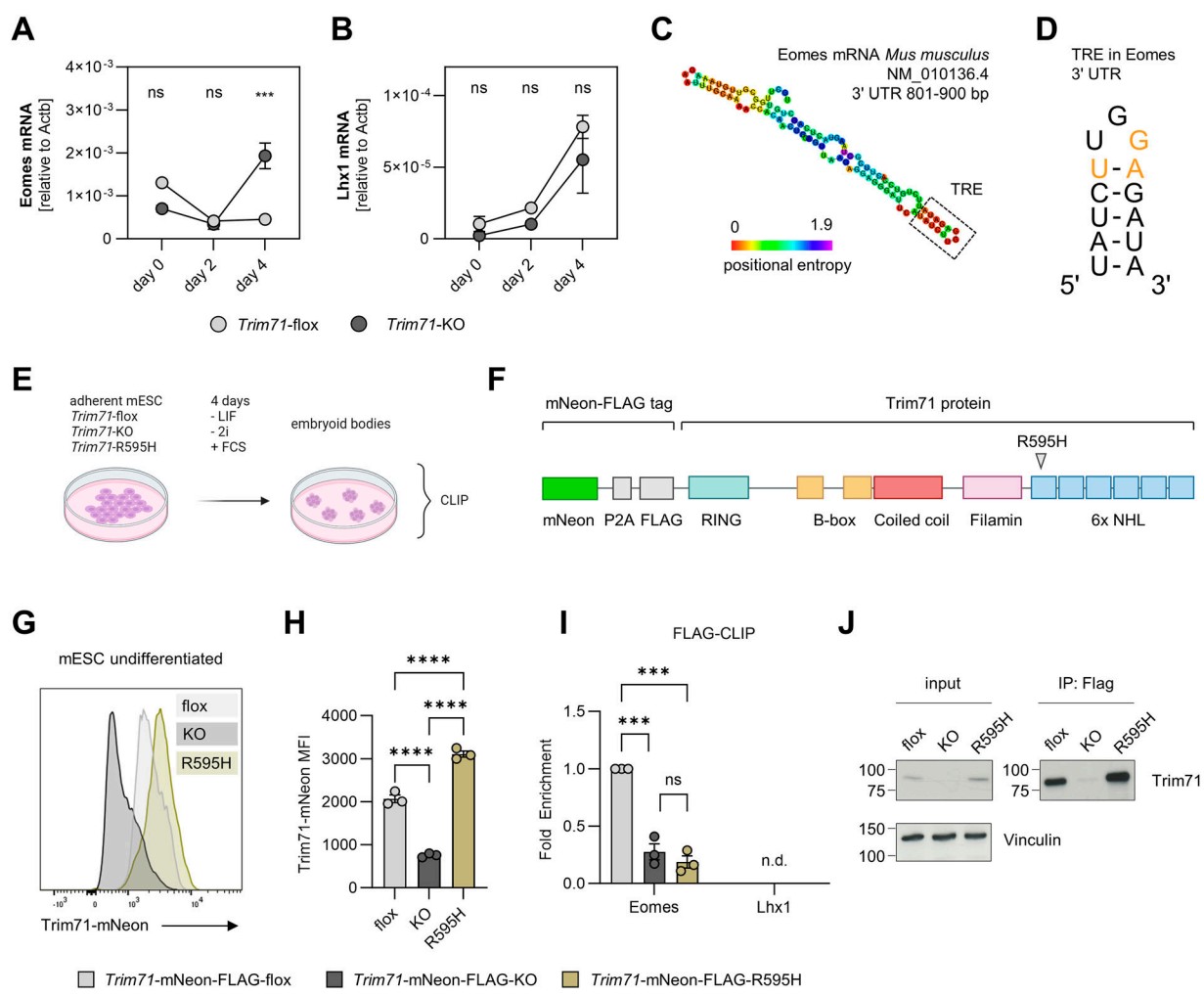

**Figure 7. Trim71 antagonizes Eomes expression by binding its mRNA via the NHL domain.**
**(A, B)** Expression of (A) Eomes and (B) Lhx1 in *Trim71*-flox and *Trim71*-KO mESC at day 0, day 2, and day 4 of differentiation as determined by qRT–PCR (n = 3; data are depicted as the mean ± SEM, two-way ANOVA, ns, not significant, ***P < 0.001). **(C)** RNAfold secondary structure prediction of the 801- to 900-bp region of the murine Eomes mRNA 3′ UTR. The identified TRE is highlighted by a dashed box. **(D)** Representation of the identified TRE in the Eomes mRNA 3′ UTR. Critical bases for the interaction with the Trim71 NHL domain are highlighted in orange. **(E)** Depiction of *Trim71*-mNeon-FLAG-WT, mNeon-FLAG-KO, or mNeon-FLAG-R595H mESC differentiation in embryoid bodies, followed by cross-linking immunoprecipitation (CLIP) at day 4. Created with BioRender.com. **(F)** Representation of the Trim71-mNeon-FLAG protein with indicated domains and position of the R595H mutation. **(G, H)** Representative histogram and (H) quantification of the median fluorescence intensity of the Trim71-mNeon signal in undifferentiated *Trim71*-mNeon-FLAG-WT, *Trim71*-mNeon-FLAG-KO, or *Trim71*-mNeon-FLAG-R595H mESC (n = 3; data are depicted as the mean ± SEM, ordinary one-way ANOVA, ****P < 0.0001). **(I)** Quantification of Eomes and Lhx1 fold enrichment in FLAG-CLIP of *Trim71*-mNeon-FLAG-WT, *Trim71*-mNeon-FLAG-KO, or *Trim71*-mNeon-FLAG-R595H mESC at day 4 of differentiation (n = 3; data are depicted as the mean ± SEM, one-way ANOVA, ns, not significant, ***P < 0.001). **(J)** Western blot of input and immunoprecipitated protein lysates after FLAG-CLIP detected with Trim71 and vinculin antibodies.

embryos is the result of both EC-intrinsic loss of angiogenic activity and the reduction of external hemodynamic forces.

As argued before, the cranial neural tube closure observed in *Trim71*-KO embryos is likely not the cause of their embryonic lethality at E9.5–E11.5, considering that embryos carrying other genetic ablations leading to neural tube defects typically survive until the late fetal period (Copp et al, 2003; Maller Schulman et al, 2008). In contrast, vascular development, primitive erythropoiesis, and proper heart function are indispensable for mid-gestational survival of the embryo, and deficiencies in either process lead to lethality at the onset of organogenesis (Shalaby et al, 1995; Fujiwara et al, 1996; Koushik et al, 2001; Coultas et al, 2005). Furthermore, the detection of defects in these processes in *Trim71*-KO embryos temporally

coincides with the appearance of developmental retardation and morphological anomalies from E9.5 on. Considering that defects in either component of the circulatory system are sufficient to drive embryonic lethality, the combined defects in all major parts of the circulatory system of *Trim71*-KO embryos provide an adequate explanation for their developmental arrest and lethality.

## Trim71-dependent control of gene expression during gastrulation determines proper vascular development and primitive erythropoiesis

We further investigated the developmental origin of the vascular and erythropoiesis defects in *Trim71*-KO embryos. Trim71 expression

was present in mesodermal progenitors, HEP, and EC, but was absent in EryP. Using *Tie2*<sup>Cre</sup> *Trim71* cKO embryos, we show that the expression of Trim71 in EC and *Tie2*-expressing progenitors of EC and EryP is dispensable for embryonic survival, heart function, and primitive erythropoiesis, and only marginally affects yolk sac angiogenesis. We therefore considered the mesoderm during gastrulation as a potential origin of *Trim71*-KO–related hematoendothelial phenotypes. At E7.5, *Trim71*-KO embryos are present at Mendelian ratios (Torres-Fernández et al, 2021) and appear normal in morphology. Mesodermal cells were also present at similar numbers in E7.5 WT and *Trim71*-KO embryos, indicating that they are normally produced during gastrulation. The presence of EC in the yolk sac of *Trim71*-KO embryos further demonstrates that mesodermal progenitors are not impeded in their migration from the primitive streak to extraembryonic sites (Saykali et al, 2019). Nevertheless, our scRNA-seq data of E7.5 embryos show that widespread transcriptional differences are induced by *Trim71*-KO already at this developmental stage, preceding the appearance of morphological phenotypes by at least 1.5 d. This suggests that molecular changes in *Trim71*-KO embryos arise at gastrulation, providing a basis for the morphological phenotypes that become apparent at organogenesis. In the mesoderm, *Trim71*-KO led to the decreased expression of genes involved in RNA splicing and chromatin remodeling. A previous study reported global changes in mRNA transcript splicing in *Trim71*-KO mESC, which was attributed to direct Trim71-mediated repression of the splicing regulator Mbnl1 (Welte et al, 2019). Although Mbnl1 was not identified as a DEG in the mesoderm in our scRNA-seq data, it is possible that Trim71 regulates RNA splicing during development by controlling the expression of multiple different splicing factors. Another recent study identified the regulation of the chromatin modifier cfp-1 by TRIM71 in *C. elegans* (Kumari et al, 2023), and the down-regulation of chromatin remodeling factors in our data could hint at the control of epigenetic regulators as an additional conserved mechanism of how Trim71 shapes gene expression programs.

Importantly, we show that *Trim71*-KO leads to an increased expression of the transcription factor Eomes in the mesoderm in vivo and in differentiated mESC in vitro. We identified a secondary structure in the 3′ UTR of the murine Eomes mRNA that fulfills the requirements of a TRE for the interaction with the NHL domain of Trim71 (Kumari et al, 2018). The location of this TRE in the Eomes mRNA is in line with the preferential positioning of TREs in the 3′ UTR of previously identified target mRNAs (Torres-Fernández et al, 2019; Welte et al, 2019; Kumari et al, 2023). The binding of Eomes mRNA by Trim71 was confirmed by CLIP and was abrogated by the R595H mutation in the RNA-binding NHL domain of Trim71 (Duy et al, 2022), establishing Eomes as a direct target of Trim71-mediated repression in mesodermal development. In contrast, Lhx1 mRNA was devoid of putative TREs and was also not detectable after Trim71-FLAG-CLIP, demonstrating that Trim71 does not directly antagonize Lhx1 expression by mRNA binding. Transcription of the *Lhx1* locus is transactivated by Eomes (Nowotschin et al, 2013); thus, the elevated Lhx1 mRNA levels in *Trim71*-KO embryos could be the result of the excessive Eomes expression in these embryos. Eomes is crucial for mesoderm formation during gastrulation (Russ et al, 2000; Arnold et al, 2008) and for the commitment of pluripotent cells to the mesodermal

and endodermal lineages by establishing the accessibility of lineage-specific enhancers (Tosic et al, 2019; Schröder et al, 2024). The regulation of Eomes mRNA levels by Trim71 therefore provides further indication that Trim71 plays a central role during gastrulation. Moreover, Eomes is required for the specification of ESC into blood progenitors, cardiac mesoderm, EryP, and yolk sac endothelium (Harland et al, 2021; Costello et al, 2011; Theeuwes et al, 2024 *Preprint*; van den Ameele et al, 2012). The differentiation of mesodermal cells into hematopoietic, cardiac, smooth muscle, or mesenchymal fate is associated with different doses of Eomes induction in ESC (Pfeiffer et al, 2018), indicating that spatiotemporal regulation of Eomes levels could be required to control the proper generation of mesoderm-derived tissues. Excessive Eomes levels in *Trim71*-KO embryos, caused by a lack of post-transcriptional repression via Trim71, might thus contribute to their decreased EryP generation and impaired cardiovascular development. Nevertheless, we cannot rule out the existence of additional Eomes-independent effects caused by the loss of Trim71 that further promote impaired cardiovascular and hematopoietic development. Such effects could be mediated by non-mesodermal tissues, considering the ubiquitous expression of Trim71 at E7.5. For example, the observed changes in cell numbers and transcriptome of the ExE endoderm in *Trim71*-KO embryos might affect blood vessel development in the yolk sac in a non–cell-autonomous manner, as previously observed in other genetic mouse models (Rhee et al, 2013). This hypothesis could experimentally be addressed with the Sox2-Cre line to target *Trim71* specifically in the epiblast (Hayashi et al, 2002) or using chimeric embryos via injection of *Trim71*-deficient mESC.

In summary, we show that defects in primitive erythropoiesis and the cardiovascular system of *Trim71*-KO embryos are not caused by the loss of Trim71 in EC, EryP, and their *Tie2*-expressing progenitors. Instead, our data strongly indicate that these phenotypes are initiated at gastrulation and manifest at the onset of organogenesis, potentially because of increased Eomes levels caused by a lack of Trim71-mediated repression of gene expression.

## Materials and Methods

### Mouse lines

All mice used in this study were bred in the licensed animal facility of the LIMES Institute (University of Bonn). All animal experiments were approved by local authorities of the state of Nordrhein-Westfalen (Landesamt für Natur, Umwelt und Verbraucherschutz NRW). The mouse lines *Trim71*-KO, *Trim71*-flox, *Csf1r*<sup>iCre</sup>, and *Tie2*<sup>Cre</sup> used in this study have been described previously (Kisanuki et al, 2001; Deng et al, 2010; Mitschka et al, 2015). Conditional knockout of *Trim71* was induced by crossing female *Trim71*<sup>fl/fl</sup> Cre–negative mice with male *Trim71*<sup>fl/fl</sup> or *Trim71*<sup>fl/+</sup> mice carrying a single copy of the Cre allele. Cre-positive *Trim71*<sup>fl/fl</sup> animals were considered as the conditional knockout group, and Cre-negative littermates with either *Trim71*<sup>fl/+</sup> or *Trim71*<sup>fl/fl</sup> genotype were used as controls. Genotyping of mice and embryos was performed as described previously (Torres-Fernández et al, 2021). The ectoplacental cone

was used for genotyping of E7.5–E8.5 embryos, whereas the embryo tail was used to genotype embryos from E9.5 on.

## Generation of mouse embryos

Mouse embryos were analyzed at developmental stages E7.5–E12.5 in this study. Timed mating was performed by placing one male and up to two female mice in a cage from afternoon until the morning of the next day. The presence of a vaginal plug was assessed as an indicator of mating, and a weight gain of more than 1.75*g* of the female mouse was regarded as a sign of actual pregnancy (Heyne et al, 2015.). Pregnant female mice were euthanized by cervical dislocation, and embryos were dissected in PBS under a SZX10 stereomicroscope (Olympus) equipped with cellSens Entry software (Olympus) for the acquisition of images and videos.

## Analysis of the heartbeat in embryos

Videos of the embryo heartbeat were obtained during the preparation process with cellSens Entry software (Olympus). Embryos that did not display any heart contractility for more than 60 s were classified as absent of a heartbeat. For embryos that showed heart contractility, at least three contractions were recorded and the heart rate was calculated as the mean duration between each contraction per minute.

## Flow cytometry and FACS

Embryonic organs (yolk sac, embryo head, embryo body) were digested (100 mg/ml collagenase D, 100 U/ml DNase I in 3% FCS/PBS) for 30 min at 37°C, diluted with 2 ml FACS buffer (2 mM EDTA, 0.5% BSA in PBS), and minced through a 100-$\mu$m strainer to generate a single-cell suspension (Iturri et al, 2017). Cells were centrifuged (320*g*, 4°C, 5 min), followed by incubation in Fc blocking solution ($\alpha$-CD16/CD32 1:200 in FACS buffer, 101302; BioLegend) for 15 min at 4°C and staining in antibody mix (all antibodies 1:200 in FACS buffer) for 30 min at 4°C. Combinations of the following antibodies were used: AA4.1 (136509; BioLegend), CD11b (101237; BioLegend), CD31 (102407, 102435; BioLegend), CD45 (748370; BD Biosciences, 103131; BioLegend), CD71 (741066; BD Biosciences), Cx3cr1 (149013; BioLegend), F4/80 (123108, 123113; BioLegend), Kit (135123, 135111; BioLegend), and TER-119 (116205, 116220, 116223). For the analysis of the Trim71-mNeon signal in undifferentiated mESC, adherent cells were detached with Accutase (Thermo Fisher Scientific) for 3 min at 37°C, followed by the addition of 2 ml FACS buffer and filtration through a 100-$\mu$m strainer. Samples were washed, resuspended in FACS buffer, and diluted 1:1 with DRAQ7 (1:1,000 in FACS buffer) or DAPI (2 $\mu$M in FACS buffer) for dead cell exclusion immediately before recording at the LSRII or FACS Symphony A5 flow cytometers (Becton Dickinson). Data analysis was performed in FlowJo (Becton Dickinson). Cell populations from embryonic organs were identified by the combination of expressed surface markers: EryP (Ter119$^+$ CD45$^-$), EC (CD31$^+$ AA4.1$^-$ in the yolk sac, CD31$^+$ in the embryo body and embryo head), EMP (CD45$^{low}$ Kit$^+$ AA4.1$^+$), pMac (CD45$^+$ Kit$^-$ CD11b$^+$ F4/80$^-$), macrophages (CD45$^+$ Kit$^-$ CD11b$^+$ F4/80$^+$). EryP were quantified as a relative percentage of all

live cells, and all other cell populations were quantified as a relative percentage of all non-erythroid (Ter119$^-$) cells.

Cell sorting was performed at ARIA III (Becton Dickinson) with a 100-$\mu$m nozzle. 1,000–5,000 cells per sample were sorted into cooled 1.5-ml reaction tubes filled with 500 $\mu$l TRIzol reagent (Thermo Fisher Scientific) and immediately transferred to −80°C until further processing.

## Immunofluorescence staining and imaging

Embryonic organs were fixed in 4% PFA/PBS for 2 h on a shaker at 4°C and washed three times with PBS. For tissue sections, embryos were incubated with 30% sucrose in PBS for 48 h, washed three times with PBS, and embedded in O.C.T. compound (Weckert Labortechnik) before freezing to −80°C. 10- to 14-$\mu$m sections were prepared at Cryostat CM3050S (Leica). Sections were dried at room temperature (rt) for 45 min, washed with PBS for 15 min, and blocked for 1 h at rt in IF blocking buffer (5% normal goat serum, 0.3% Triton X-100, 0.5% BSA in PBS). Sections were stained with primary antibody mix (all antibodies 1:200 in PBS) overnight at 4°C. Sections were then washed three times with PBS and incubated in secondary antibody mix (all antibodies 1:400 in PBS) including 1 ng/$\mu$l DAPI for 1 h at rt. After washing three times with PBS, sections were mounted with Fluoroshield (ImmunoBioScience). Yolk sacs were stained based on a published protocol (Roy & Delgado-Olguin, 2018) as whole mounts floating in 48-well plates with one organ per well. The blocking and staining procedure for yolk sacs was the same as described for tissue sections, but incubation steps were performed on a shaker and the secondary antibody staining was extended to 90 min. The primary antibodies used for immunofluorescence staining were against CD31 (MA3105; Thermo Fisher Scientific), Flk1 (752945; BioLegend), and Trim71 (Worringer et al, 2014), which were detected with species-matched secondary antibodies anti-hamster Cy3 (127-165-160; Jackson ImmunoResearch), anti-rat AF555 (A-21434; Thermo Fisher Scientific), and anti-rabbit AF488 (A-11034; Thermo Fisher Scientific). Images were acquired using a LSM 880 Airyscan confocal microscope (Carl Zeiss). Image analysis was performed in ZEN blue (Carl Zeiss) and ImageJ. For the quantification of vascular structures in the yolk sac, three images per sample from different locations in the yolk sac were analyzed and the mean value was used as one data point. Endothelial extensions were defined as CD31$^+$ structures that extend from an existing vessel without any connection to another vessel. Branching points were defined as the intersection of at least three vessels.

## Stem cell culture and differentiation

The *Trim71*-flox and *Trim71*-KO mESC lines, and the mNeon-FLAG-tagged *Trim71*-flox, *Trim71*-KO, and *Trim71*-R595H mESC lines have been described previously (Mitschka et al, 2015; Duy et al, 2022). Embryonic stem cells were maintained at 37°C, 5% CO$_2$, and 95% relative humidity on gelatin-coated dishes in KnockOut DMEM (Thermo Fisher Scientific) supplemented with 15% FCS (Thermo Fisher Scientific), penicillin–streptomycin (Thermo Fisher Scientific), GlutaMAX (Thermo Fisher Scientific), MEM Non-essential Amino Acids (Thermo Fisher Scientific), 50 $\mu$M $\beta$-mercaptoethanol (Thermo Fisher Scientific), LIF (supernatant from L929 cells) and 2i

CHIR99021 (3 $\mu$M; Sigma-Aldrich), and PD0325901 (1 $\mu$M; StemCell Technologies).

Differentiation of mESC was performed as embryoid bodies (EBs) in suspension by the removal of 2i and culturing in differentiation medium: StemPro-34 SFM (Thermo Fisher Scientific) supplemented with 10% FCS, penicillin–streptomycin, 2 mM L-glutamine (PAN-Biotech), 40 $\mu$g/ml apotransferrin (Merck), 0.5 mM L-ascorbic acid (Sigma-Aldrich), and 0.15 mM 1-MTG (Sigma-Aldrich). To this end, adherent mESC were detached with Accutase (Thermo Fisher Scientific) and washed twice with PBS, and $2.5 \times 10^6$ cells in 5 ml differentiation medium were seeded into a 60-mm non-treated petri dish (Greiner Bio-One). Cells were differentiated for 4 d with a medium change after 2 d without disruption of the EBs.

### Analysis of gene expression by qRT–PCR

RNA was extracted using TRIzol reagent (Thermo Fisher Scientific) according to the manufacturer's instructions. For low input samples, linear polyacrylamide (LPA) was added to the sample before RNA extraction (2.5 ng LPA for FACS-isolated cells, 7.5 ng LPA for CLIP samples). Isolated RNA was digested with 1 U/$\mu$l DNase I (Thermo Fisher Scientific) for 30 min at 37°C, followed by the addition of 1 $\mu$l 50 mM EDTA and heat inactivation for 10 min at 70°C. 500 ng RNA was reverse-transcribed using High-Capacity cDNA Reverse Transcription Kit (Thermo Fisher Scientific) according to the manufacturer's instructions. Relative gene expression was quantified by qRT–PCR with TaqMan or SYBR Green assays (Bio-Rad) on a CFX96 thermal cycler (Bio-Rad). The following TaqMan probes were used: Trim71 (Mm01341471_m1), Eomes (Mm01351985_m1). For SYBR Green assays, the following primer pairs were used:

18S (for: GTAACCCGTTGAACCCCATTC, rev: CCATCCAATCGGTAGTAGCGAC).

Actb (for: CACTGTCGAGTCGCGTCC, rev: CGCAGCGATATCGTCATCCA).

Flk1 (for: TAGCTGTCGCTCTGTGGTTC, rev: TTCTGTGTGCTGAGCTTGGG).

Lhx1 (for: CGCCATATCCGTGAGCAACT, rev: CGCGCTTAGCTGTTTCATCC).

Pdgfra (for: GAGATCGAAGGCAGGCACAT, rev: GGCAGAGTCATCCTCTTCCAC).

### CLIP

CLIP was performed as described previously (Torres-Fernández et al, 2019). Briefly, for each condition the EBs from six 60-mm dishes of mESC expressing FLAG-tagged Trim71 variants at day 4 of differentiation were pooled, washed with cold PBS, and UV-irradiated at 254 nm and 300 mJ/cm². After centrifugation (320$g$, 4°C, 5 min), cells were lysed in TKM+ buffer for 15 min on ice (20 mM Tris, 100 mM KCl, 5 mM MgCl$_2$, pH 7.4, supplemented with protease inhibitors 1:1,000 PMSF, 1:1,000 benzamidine, 1:1,000 antipain; 1:2,000 aprotinin, 1:2,000 leupeptin, and 0.2% NP-40, and RNase inhibitor 120 U/ml). Lysates were centrifuged (16,100$g$, 4°C, 5 min), and a fraction of the supernatant was retained for protein and RNA input fractions. Equal amounts of protein from the supernatant (500–1,500 mg) were immunoprecipitated with 30 $\mu$l anti-FLAG M2 magnetic beads (Sigma-Aldrich) at 4°C on a spinning wheel for 4 h. After washing five times with TKM+ buffer, 20% of the IP fraction was used for Western blot

analysis, whereas 80% were digested with 0.5 mg/ml Proteinase K at 37°C for 30 min and used for RNA extraction, cDNA generation, and qRT–PCR analysis of target genes and 18S RNA as an unspecific binding reference gene. Enrichment values were calculated as enrichment = $2^{\wedge} - [(Ct_{CLIP\_target} - Ct_{CLIP\_ref}) - (Ct_{Input\_target} - Ct_{Input\_ref})]$ and normalized to the control genotype. Input and IP protein fractions were analyzed by Western blot as described previously (Torres-Fernández et al, 2019). Briefly, protein lysates were separated by size via SDS–PAGE and transferred to a nitrocellulose membrane. Membranes were incubated overnight at 4°C with Trim71 (Worringer et al, 2014) or vinculin (V9131; Sigma-Aldrich) antibodies (both 1:1,000 in 5% milk powder in TBST: 50 mM Tris–HCl, pH 7.6, 150 mM NaCl, 0.05% Tween-20), washed three times with TBST, and incubated with species-matched HRP-coupled secondary antibodies (7074 or 7076 at 1:5,000 in 5% milk powder in TBST; Cell Signaling Technology) for 1 h at rt. After washing three times with TBST, membranes were developed with Pierce ECL Substrate Kit (Thermo Fisher Scientific).

### Single-cell RNA sequencing (scRNA-seq)

The 10x Genomics Chromium Next GEM Single Cell 3′ Reagent Kits v3.1 (Dual Index) kit was used for scRNA-seq experiments. For developmental stage E7.5, one whole embryo of each genotype was used for sequencing. Single cells were isolated by digestion of E7.5 embryos with 0.25% Trypsin (Sigma-Aldrich) and 0.5 mM EDTA in PBS for 10 min at 37°C, followed by mechanical dissociation by gentle resuspension through a 200-$\mu$l pipette tip and filtering through a 40-$\mu$m strainer into a 1.5-ml reaction tube. For E9.5 stage yolk sacs, cells isolated from two organs per genotype were pooled into one sample. Yolk sacs were digested with 100 mg/ml collagenase D and 100 U/ml DNase I in 3% FCS/PBS for 30 min at 37°C, diluted to 1.5 ml with PBS, minced through a 100-$\mu$m pore strainer, and filtered through a 70-$\mu$m pore strainer into a 1.5-ml reaction tube. Cells isolated from E7.5 whole embryos or E9.5 yolk sacs in 1.5-ml reaction tubes were centrifuged (400$g$, 4°C, 5 min) and completely loaded onto Next GEM Chip G. Subsequent sample cleanup and library preparation were performed according to the manufacturer's instructions. Libraries were sequenced on NovaSeq 6000 System (Illumina) with paired-end dual indexing (28 cycles Read 1, 10 cycles i7, 10 cycles i5, 90 cycles Read 2) with NovaSeq 6000 S2 and SP (200 cycles) chemistry.

### Analysis of scRNA-seq data

Data generated in this study were processed using Cell Ranger v7.1.0 (10x Genomics). Specifically, raw sequencing data were demultiplexed with the cellranger mkfastq pipeline. The generated FASTQ files were further processed using the cellranger count pipeline for alignment, filtering, barcode counting, UMI counting, and the generation of feature–barcode matrices. Mm10 2020A was used as a mouse reference genome. Subsequent data analysis was performed in R using Seurat (v.5.0) (Hao et al, 2024). Ambient RNA was removed by SoupX (Young & Behjati, 2020), and high-quality cells with 500–5,000 features and less than 5% mitochondrial reads were filtered. The datasets of both genotypes were merged, normalized, and scaled with standard settings from the Seurat package, followed by dimensionality reduction and visualization by UMAP. Cell types were annotated with SingleR and a reference dataset that was

accessed via the MouseGastrulationData package (Pijuan-Sala et al, 2019). Cells from E7.5 embryos provided by this reference dataset were used to identify cell types in the E7.5 whole-embryo scRNA-seq experiment. For the E9.5 yolk sac scRNA-seq experiment, cell-type annotations from E8.5 embryos were used as a reference (Pijuan-Sala et al, 2019). Hematopoietic cell populations in the yolk sac were further discriminated based on reported gene expression signatures for EMP, pMac, and macrophages (Mass et al, 2016), as well as defined expression thresholds for *Maf* (macrophages) and *Pf4* (megakaryocytes). Differentially expressed genes between genotypes were identified for each cell population using the FindMarkers function from Seurat (Wilcoxon's rank sum test), and filtered for an adjusted $P$-value of < 0.05 and an expression fold change of >1.2 (up-regulated) or <0.8 (down-regulated). Up- or down-regulated DEG were used for gene ontology overrepresentation analysis with the clusterProfiler package, accessing MSigDB gene annotations (Wu et al, 2021; Castanza et al, 2023). Expression scores for selected MSigDB processes were calculated with the AddModuleScore function from Seurat.

ScRNA-seq data of WT E6.5–E8.5 embryos were retrieved from the mouse gastrulation and early organogenesis cell atlas (Pijuan-Sala et al, 2019). Pseudo-bulk expression data were generated by accessing the data of embryonic stage E7.5 and E8.5 cells via the MouseGastrulationData package in R and using the AggregateExpression function from Seurat. Cells annotated as Erythroid3 in the source dataset were considered as EryP in the analysis.

### Prediction of RNA secondary structure

The sequence of the murine Eomes mRNA was downloaded from the NCBI (accession: NM_010136.4) and divided into 5′ UTR, coding sequence, and 3′ UTR. Each region was further divided into 100-bp fragments that were individually analyzed with the RNAfold tool (http://rna.tbi.univie.ac.at/cgi-bin/RNAWebSuite/RNAfold.cgi) and displayed with color-coded positional entropy of each nucleotide.

# Data Availability

The scRNA-seq data were submitted to the NCBI Gene Expression Omnibus with the accession number GSE272044. For both scRNA-seq experiments reported by this study, DEG in between *Trim71*-WT and *Trim71*-KO genotypes for each cell population are available in Supplemental Data 1 and Supplemental Data 2.

# Supplementary Information

# Acknowledgements

We thank Andrea Raths, Cornelia Cygon, and Jordi Hees Soler for technical support. We are grateful to Shinya Yamanaka for providing the Trim71 antibody used in this study. Moreover, we are very thankful to Carmen Ruiz de Almodóvar for feedback on this article. The work was funded by the Deutsche Forschungsgemeinschaft (DFG, German Research Foundation) under Germany's Excellence Strategy-EXC2151-390873048 (to W Kolanus, E Mass, M Beyer, and A Schlitzer); SFB 1454—Project ID 432325352 (to W Kolanus, E Mass, A Schlitzer, and M Beyer); FOR5547—Project ID 503306912 (to E Mass); and the Bonner Forum Biomedizin (BFB Program for PhD students supporting innovative ideas, to T Beckröge). E Mass is supported by the European Research Council (ERC) under the European Union's Horizon 2020 research and innovation program (Grant Agreement No. 851257). We would like to thank the Flow Cytometry Core Facility—Campus Poppelsdorf—for providing support and instrumentation.

## Author Contributions

T Beckröge: conceptualization, formal analysis, investigation, methodology, and writing—original draft.
B Jux: investigation and methodology.
H Seifert: investigation and methodology.
H Theobald: investigation and methodology.
E De Domenico: formal analysis and investigation.
S Paulusch: formal analysis and investigation.
M Beyer: investigation and methodology.
A Schlitzer: investigation.
E Mass: investigation and methodology.
W Kolanus: conceptualization, funding acquisition, investigation, methodology, and writing—original draft, review, and editing.

## Conflict of Interest Statement

The authors declare that they have no conflict of interest.

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
