## [Reviewer comments · Life Science Alliance]

Life Science Alliance

Impaired primitive erythropoiesis and defective vascular development in Trim71-KO embryos

Tobias Beckröge, Bettina Jux, Hannah Seifert, Hannah Theobald, Elena De Domenico, Stefan Paulusch, Marc Beyer, Andreas Schlitzer, Elvira Mass, and Waldemar Kolanus

DOI: <https://doi.org/10.26508/lsa.202402956>

Corresponding author(s): Waldemar Kolanus, University of Bonn

Review Timeline:

Submission Date:	2024-07-24
Editorial Decision:	2024-09-04
Revision Received:	2025-01-17
Editorial Decision:	2025-01-21
Revision Received:	2025-01-27
Accepted:	2025-01-27

Transaction Report:

September 4, 2024

Re: Life Science Alliance manuscript #LSA-2024-02956-T

Prof. Waldemar Kolanus
University of Bonn
LIMES Program Unit Molecular Immune and Cell Biology, Laboratory of Molecular Immunology
Carl-Troll-Straße 31, Bonn
Bonn, NRW 53115
Germany

Dear Dr. Kolanus,

Thank you for submitting your manuscript entitled "Dysregulation of gene expression during gastrulation results in impaired primitive erythropoiesis and vascular development in Trim71-KO embryos" to Life Science Alliance. The manuscript was assessed by expert reviewers, whose comments are appended to this letter. We invite you to submit a revised manuscript addressing the Reviewer comments.

Thank you for this interesting contribution to Life Science Alliance. We are looking forward to receiving your revised manuscript.

Sincerely,

B. MANUSCRIPT ORGANIZATION AND FORMATTING:

Reviewer #1 (Comments to the Authors (Required)):

This study by Beckröge and collaborators reports the role of Trim71 in mesoderm specification to the cardio-vascular system during early embryonic development. This is a very well-conducted study that convincingly demonstrates how Trim71 controls EOMES levels in mesoderm subsets. The experimental strategy allows an in-depth characterisation of trim71 role in early cardio-vascular development making use of conditional knock-outs and scRNA-seq very effectively. Overall, all the data presented are very supportive of the claims and conclusions.

Several points should be addressed by the authors:

In the introduction and throughout the manuscript, it should be made clear that hemogenic endothelial cells (HEC) are the progenitors giving rise to blood cells. HEC do not give rise to endothelial cells. Hematoendothelial progenitors (HEP) is an ill-defined population (is that the hemangioblast population?), and this terminology should not be used.

"Transient definitive hematopoietic cells also emerge in the yolk sac by the differentiation of EC into erythro-myeloid progenitors (EMP) at E8.5": this sentence is incorrect, it should be "the differentiation of HEC"

Page 6: "Altogether, these data show that Trim71-KO leads to a defect in primitive erythropoiesis." It is a decrease rather than a defect.

The immunophenotypes for EC (supp figS1D-F0), EMP and pMac (Fig 3) should be described in the main text or in the figure legend.

Page 7: "We analyzed the translocation efficiency of EMP-derived pMac from the yolk sac to the embryo proper". Translocation is not the most appropriate terminology for this process, migration might be more suitable.

It is not clear if Csf1^{ricre} Trim71 animals have any specific phenotype. This should be mentioned in the text.

Why is Trim71 expression in the yolk sac so punctate (supp fig 4C), as compared to the other tissues (supp fig 4B, D)?

Why are there large differences in some cell populations (supp Fig 5C)?
e.g. primitive streak or ExE

Reviewer #2 (Comments to the Authors (Required)):

In this paper, the authors investigate the function of the RNA-binding protein Trim71 during early murine embryogenesis. This paper is very well-written and the data are logically laid out and presented. The author show that Trim71 knockout embryos die at E10.5-E11.5 with significant defects of the cardiovascular system, including a mark reduction of primitive erythroid cells, vascular remodeling defects in the yolk, sac, and a decrease in cardiac function (contractions and heart rate). These defects are in addition to the previously described neuronal and germ cell defects, and establish the likely cause of lethality in embryos lacking Trim71.

Consistent with defects in the cardiovascular system, markedly reduced migration of macrophage cells was noted into the head and remainder of the embryo body. Normal numbers of macrophage lineage cells and "EMP" were present in the yolk sac at E9.5 and E10.5 conceptuses. Consistent with these findings, functional deletion of Trim71 in erythro-myeloid progenitors/macrophage lineage cells using the inducible Csf1^{rCre} mouse model failed to elicit defects in those hematopoietic lineages. Single cell RNA-Seq studies of E9.5 yolk sacs in wild-type and Trim71-null embryos confirmed the marked reduction in primitive erythroid cells and revealed gene expression differences in endothelial cells.

Since hematopoietic and (some, but not all) endothelial cells are thought to arise from common progenitors, Tie2Cre was used to ablate Trim71 from these cell types. Surprisingly, there were only mild endothelial defects and no evidence of functional cardiac or primitive erythroid defects in Trim71-targeted embryos. Given these results, indicating that Trim71 is not directly responsible for the hematopoietic and cardiac defects noted in the KO embryo, the investigators postulated that Trim71 must be functioning earlier in development and examined gastrulating mouse embryos at E7.5 by scRNA-Seq.

Single cell RNA-Seq studies at E7.5 were conducted to better understand the etiology of the cardiovascular defects. Decreases in "mesenchyme" and "hemato-endothelial progenitors" were noted. While not specifically described, it is interesting that the number of extraembryonic endoderm cells is markedly expanded (sFig. 5C) and their position in UMAP space is different (Fig. 6A). Since extraembryonic endoderm has been described as impacting vascular and hematopoietic development in the yolk sac (e.g., Miura, 1969; Palis, 1995; Bielinska, 1996; Belaousoff, 1999), an analysis of gene differences in this tissue should be undertaken.

The single cell RNA-Seq analysis of 'mesodermal progenitors' leads to the identification of Eomes as a potential target of Trim71. Elegant mechanistic studies confirm that Eomes is a direct target of Trim71 binding and the high expression of Eomes in Trim71-null mesoderm is consistent with the known repressive function of Trim71. The authors suggest that altered expression of Eomes contributes to the defects in cardiovascular development, i.e., the defects in yolk sac endothelial cells, in primitive erythropoiesis, and the cardiac defects. While not necessary, the paper would be strengthened by evidence that overexpression of Eomes leads to one or more of these defects in differentiating mES cells.

Finally, the discussion very nicely brings up multiple excellent points, including the importance of flow for vascular remodeling of the yolk sac and the importance of Eomes in the emergence of primitive erythroid cells in the yolk sac.

A final issue should be considered:

The expression of Trim71 shown in Supplemental Figure 4B-D raises some questions. It would appear from sFig. 4B, that Trim71 is most highly expressed in extraembryonic (visceral) endoderm. Is that correct? If so, was that high expression evident in the transcriptomic data from Pijuan-Sala, et al.? If so, those data should be shown and the possibility that extraembryonic endoderm impacting vascular and hematopoietic development in the yolk sac entertained/discussed (as mentioned above). Additionally, in sFig. 4C, it is not clear to me that Trim71 is expressed in CD31+ cells at 9.5. The small green 'dashes' of positivity seems very different from the diffuse staining in sfigures B and D.

Reviewer #3 (Comments to the Authors (Required)):

1. A short summary of the paper, including description of the advance offered to the field.

In this study, Beckroge et al. demonstrate that disrupting Trim71 function during mouse embryogenesis leads to cardiovascular defects, which likely underpins the embryonic lethality observed in mutant embryos midway through gestation. The researchers identify noticeable defects beginning at embryonic day 9.5 (E9.5), including pronounced developmental delays in Trim71 knockout (KO) embryos. These delays are associated with vascular remodelling defects, the absence of major vitelline vessels in the yolk sac, a reduction in primitive erythrocytes, and impaired cardiac function. To pinpoint the timing of Trim71's role in yolk sac hematopoiesis and vascular development, they specifically disrupted Trim71 in hematoendothelial progenitors (using Tie2-Cre) and in erythro-myeloid progenitors (using Csf1r-Cre). The conditional knockout studies revealed that Trim71 expression in these progenitors is not the primary cause of the disruptions observed in Trim71 KO embryos. Through single-cell RNA sequencing of E7.5 Trim71 KO embryos, the team identified upregulation of the transcription factor Eomes in mesoderm and endoderm cells. Further investigation using an embryonic stem cell model of mesoderm formation showed that Trim71 directly regulates Eomes expression, as demonstrated by immunoprecipitation experiments. Collectively, this work establishes Trim71 as a key regulator of Eomes mRNA expression during early stages of murine gastrulation and underscores its crucial role in early yolk sac and cardiac development in the mouse embryo.

2. For each main point of the paper, please indicate if the data are strongly supportive. If not, explicitly state the additional experiments essential to support the claims made and the timeframe that these would require.

Main point #1: Trim71 is required for primitive erythropoiesis and cardiovascular development

The morphological analyses of developing mutant embryos via flow cytometry and fluorescence imaging strongly support the conclusions that Trim71 is required for normal cardiovascular development and primitive erythropoiesis.

Main point #2: Trim71-dependent control of gene expression during gastrulation determines proper vascular development and primitive erythropoiesis

The researchers demonstrate that Trim71 plays a critical role in yolk sac (YS) vascular development, primitive erythropoiesis,

and heart development. They suggest that the observed defects likely stem from disruptions in Trim71-dependent gene expression during early mesoderm formation, particularly through its regulation of Eomes expression. Given that Trim71 is broadly expressed in various early embryonic cell types beyond the mesoderm, and considering that conditional disruption of Trim71 in the epiblast (for example, using a Sox2-Cre mouse line or by generating chimeras with Trim71 KO embryonic stem cells) has not been performed, the authors should acknowledge the possibility that these defects might arise from non-cell-autonomous roles of Trim71, such as in the visceral endoderm, which, along with the extraembryonic mesoderm, forms the visceral yolk sac.

While additional conditional knockout experiments could provide further insights, they are not strictly necessary, though they would be valuable. Instead, the authors should emphasize the number of differentially expressed genes identified in other cell types in their single-cell RNA sequencing (scRNA-seq) analyses at E7.5/E9.5. Currently, the focus is primarily on the endothelium and mesoderm, while the rationale for this focus is obvious, the authors should explore and outline gene expression changes in other cell types, such as the visceral endoderm. These additional analyses could help rule out non-cell-autonomous roles of Trim71 that might contribute to YS formation defects. For instance, if a significant number of differentially expressed genes are found in the visceral endoderm at E9.5, these changes could have induced alterations in gene expression within the Trim71 mutant endothelium, potentially underlying the observed vascular defects in the YS.

The Trim71 KO embryos exhibit developmental delays, which adds another layer of complexity to the interpretation of the data. It would be valuable for the researchers to integrate their findings with the scRNA-seq data presented in Imaz-Rosshandler (2024) from E8.5 to E9.5 YS sections. This integration could help assess whether the mutant endothelium aligns with a delayed differentiation trajectory, potentially explaining many of the differentially expressed genes identified in their study. Understanding whether the observed gene expression changes are a consequence of delayed development, rather than direct effects of Trim71 loss, could provide important insights into the mechanisms underlying the vascular defects in Trim71 KO embryos.

Finally, it is worth noting that Eomes is not only crucial for the formation of primitive erythrocytes but also for the development of cardiac mesoderm and heart formation, as shown by Costello et al. (2011). Given that Trim71 KO embryos exhibit cardiac function defects, the authors should reference this study to strengthen their discussion.

3. Lastly, indicate any additional issues you feel should be addressed (text changes, data presentation, statistics etc.).

Figure 3

- Which markers were used to determine EMP (CD41, cKit, AA4.1?) and pMac percentages is not clearly presented in the text or figure. While this information is provided in the methods section, it would be beneficial to also include it in the main text for clarity and better accessibility.

Figure 4

- The scRNA-seq analysis of the yolk sac at E9.5 should include a comparison of the molecular properties of the cells with the gastrulation atlas (Imaz-Rosshandler 2024) to investigate whether the observed transcriptional disruptions are linked to a developmental delay. Specifically, this comparison could reveal if the endothelium in the Trim71 KO embryos is less mature, mapping to earlier developmental stages, such as E8.5. The decreased expression of Cdh5 observed in the data supports this possibility. Furthermore, many of the DEGs might be due to the block in maturation.

- What about the transcriptional changes in other cell types beyond the endothelium? Given that Trim71 is broadly expressed, it may also play a role in the extraembryonic endoderm lineage, which is important for primitive erythropoiesis. This could potentially explain some of the defects observed in the mutants. The authors should at least address this possibility in the discussion. Additionally, it would be helpful to plot differentially expressed genes (DEGs) for each cell type, including primitive erythrocytes, to provide a more comprehensive view of Trim71's impact across various lineages. This should be included in supplement.

SFig4

- Is there strong expression in the visceral endoderm? I

- It would be helpful to include a control or at least indicate what is considered a positive signal in the images, perhaps using arrowheads for clarity. Additionally, the staining pattern in the E9.5 yolk sac appears noticeably different from that in the gastrula and embryo. It would be valuable to address this discrepancy and provide an explanation for the observed differences.

SFig5

- A, include the genotype label in the heatmap

Figure 5

- Would be nice to see the number of DEGs per cell type, to help with interpretation of what underlies the defects in YS hematopoiesis

- Should distinguish definitive endoderm in the labels of the cell types from the visceral endoderm (ExE)

Figure 6.

- In the discussion, the authors propose that the overexpression of Eomes in the mesoderm of Trim71 KO embryos might

contribute to the observed reduction in primitive erythropoiesis. Notably, a recent preprint study (<https://www.biorxiv.org/content/10.1101/2024.08.13.607790v1>) demonstrates that Eomes knockout blocks yolk sac formation both in vitro and in vivo, suggesting that the reduction in primitive erythropoiesis observed in Harland et al., 2021, may be due to an earlier loss of yolk sac-fated mesoderm. Additionally, the discussion should reference further studies that have shown Eomes induction in embryonic stem cell (ESC) models, in various signaling contexts and at different doses, promotes the formation of lineages where Eomes is normally required, such as definitive endoderm and cardiac mesoderm (<https://www.ncbi.nlm.nih.gov/pmc/articles/PMC3321156/>, <https://www.nature.com/articles/s41467-017-02812-6>). These references would provide a broader context for understanding the role of Eomes in lineage specification.

Figure 7

- What cell types are being generated in the embryoid bodies (EBs)-endoderm, mesoderm?
- It would have been informative to stain for Flk1/PdgfRa and assess the formation of primitive erythrocytes (EryP) later in differentiation, using the flow cytometry markers mentioned in the paper.
- Since Eomes is expressed in the anterior primitive streak during the formation of definitive endoderm and cardiac mesoderm, it would be valuable to clarify the specific contexts in which these critical molecular experiments were conducted.
- Why do you think Lhx1 expression remains unchanged, despite it being a direct Eomes target? Is it possible that you examined it too early in the developmental process?

Reviewer #4 (Comments to the Authors (Required)):

In this study, Beckröge and colleagues revealed that Trim71 KO embryos show deficiency in primitive erythropoiesis, yolk sac vasculature, heart function and circulation. Trim71 deletion cause transcriptional change in mRNA processing and RNA splicing related genes in the mesoderm at E7.5. Interestingly, Trim71 KO show elevated Eomes expression not only in the mesoderm, but also endoderm. This involves the direct interaction between Trim71 and Eomes mRNA. While these findings are of interest, there is additional experiments needed to confirm some of findings.

Main comments

- Fig. 1A: It is unclear from the bright field images if the formation of the primitive streak in Trim71 KO embryos is impaired or not. Sectioning and H&E staining or whole mount staining for Brachyury/T or other primitive streak marker gene will be required to show if primitive streak formation is normal in Trim71 deletion.
- Fig. 1A and B: The authors mentioned that Trim71 KO embryos appear pale in colour from E9.5 onwards, but this was not observed in a previous study (Mitschka et al., 2015). Is this due to using a different genetic background of Trim71 mutants, although the authors used the same mouse lines as Mitschka et al? What is the difference from Mitschka et al?
- Supplemental Fig. S1D-F: The quantification method for the relative number of ECs is unclear. The methods section defines ECs as CD31+/AA4.1- cells, yet Ter119- cells are counted as ECs. This discrepancy should be clarified in the methods section.
- Fig 2: The manuscript does not clearly address whether heart development in Trim71 KO embryos is normal. The authors should investigate if the reduced heart beat rate is linked to developmental defects in the heart, and whether cardiac abnormalities and circulatory issues could contribute to embryonic lethality.
- Fig. 4A: The population for EC looks small in the scRNA-seq data. Is this consistent with the flow data for EC population in the yolk sac of E9.5?
- Regarding scRNA-seq for E7.5 embryos, one whole embryo of each genotype is not sufficient to detect biological effects of Trim71 deletion. Given the variability and different developmental speed within the same litters, more replicates (at least two, preferably more than three) are required to accurately represent gene expression profiles in Trim71 KO embryos.
- Fig. 7A and B: Are the differentiation kinetics, such as the emergence of Flk1 high cells into mesodermal lineages, equivalent in Trim71 KO compared to WT?
- Does Trim71 R595H ESC show upregulation of Eomes mRNA when differentiated to embryoid bodies and mesoderm lineages for 4 days, similar to Trim71 KO? Also, does Trim71 R595H mutant ESC show defects in mesoderm and erythroid differentiation? It is unclear if the misinteraction with Eomes in Trim71 deletion causes erythroid differentiation defects in the mutant embryos.

Minor comments

- Page 5: "characteristic embryonic morphology until E9.5" - As E9.5 Trim71 KO embryos show size defects and cranial neural tube closure defects, Trim71 KO embryos acquired the characteristic morphology until E8.5 and showed defects afterwards.
- Fig. 1C: EryP is analyzed by Ter119 and CD71 expression by flow cytometry in many other studies. Can the author examine CD71 /Ter119 in the E10.5 yolk sac?
- Supplementary Fig. 2B and C: Does MFI mean "Mean Fluorescence Intensity"?
- Fig. 3 and supplemental Fig. S1D-F: All cell types are shown as "% of Ter119- cells". Please correct this if these are mislabeled.
- Fig. 6B: The use of light gray and dark gray in the UMAP plot makes it hard to see the difference between WT and Trim71 KO cells. A more distinct colour combination is recommended to better illustrate the distribution changes of Trim71 KO cells.
- Is Trim71 deletion by Tie2 Cre in conditional knockout embryos confirmed by qPCR or IF? What was the efficiency of Trim71 deletion in the conditional knockout?
- The details of antibodies used in this study should be listed.

Response to Reviewers

Reviewer #1

This study by Beckröge and collaborators reports the role of Trim71 in mesoderm specification to the cardio-vascular system during early embryonic development. This is a very well-conducted study that convincingly demonstrates how Trim71 controls EOMES levels in mesoderm subsets. The experimental strategy allows an in-depth characterisation of trim71 role in early cardio-vascular development making use of conditional knock-outs and scRNA-seq very effectively. Overall, all the data presented are very supportive of the claims and conclusions.

Several points should be addressed by the authors:

Q1.1: In the introduction and throughout the manuscript, it should be made clear that hemogenic endothelial cells (HEC) are the progenitors giving rise to blood cells. HEC do not give rise to endothelial cells.

A1.1: We have corrected the text passage accordingly (p. 4).

Q1.2: Hematoendothelial progenitors (HEP) is an ill-defined population (is that the hemangioblast population?), and this terminology should not be used.

A1.2: There is currently no consistent terminology of this cell population within the literature. The term “Hematoendothelial progenitors (HEP)” in our study is based on the terminology used by several recent single cell RNA seq studies (Pijuan-Sala et al. 2019; Biben et al. 2023; Imaz-Rosshandler et al. 2024). We chose not to use the “hemangioblast” terminology, since there are other intermediary mesoderm-derived cell populations (e.g. mesenchymo-angioblast and haematomesoblast) that also contribute to the EC and EryP pool of the embryo, respectively (Biben et al. 2023). In our view, “HEP” appears to be the most inclusive terminology, but we acknowledge its limitations, e.g., the lack of a precise spatial definition of cells included in this population within the developing embryo. For the scRNA sequencing data presented in our study (datasets generated by us that were annotated using Pijuan-Sala et al. 2019 as reference and re-analyzed expression data from Pijuan-Sala et al. 2019), we kept the “HEP” terminology to maintain consistency with the source study of Pijuan-Sala et al. 2019. However, we modified the introduction section to make more clear where this term originates from (p. 4). For the results and discussion sections on the *Tie2^{Cre} Trim71* cKO line, we now no longer refer to *Tie2^{Cre}* targeted cells as HEP, since the progenitors of EC and EryP that are targeted via the *Tie2^{Cre}* line has to our knowledge not yet been connected to the “HEP” nomenclature applied by scRNA-seq studies (p. 2 abstract, p. 5, p. 11, p. 17, p. 19). We hope that the referee will agree with our reasoning for this choice.

Q1.3: "Transient definitive hematopoietic cells also emerge in the yolk sac by the differentiation of EC into erythro-myeloid progenitors (EMP) at E8.5": this sentence is incorrect, it should be "the differentiation of HEC"

A1.3: We have changed this sentence to: “Transient definitive hematopoietic cells also emerge in the yolk sac by the differentiation of hemogenic endothelial cells (HEC) into erythro-myeloid progenitors (EMP) at E8.5” (p. 4).

Q1.4: Page 6: "Altogether, these data show that Trim71-KO leads to a defect in primitive erythropoiesis." It is a decrease rather than a defect.

A1.4: We have changed the respective sentence to: "Altogether, these data show that Trim71-KO leads to a decrease in primitive erythropoiesis." (p. 6).

Q1.5: *The immunophenotypes for EC (supp fig S1D-F0), EMP and pMac (Fig 3) should be described in the main text or in the figure legend.*

A1.5: We have added the surface markers which are used to define cell types by flow cytometry (EryP, EC, EMP, pMac, Macrophages) to the results section of the main text (p. 6-8).

Q1.6: *Page 7: "We analyzed the translocation efficiency of EMP-derived pMac from the yolk sac to the embryo proper". Translocation is not the most appropriate terminology for this process, migration might be more suitable.*

A1.6: We have replaced the word "translocation" with the term "migration" (p. 7+8, p. 14).

Q1.7: *It is not clear if Csf1ricre Trim71 animals have any specific phenotype. This should be mentioned in the text.*

A1.7: We have added genotype quantifications of litters and light microscopy pictures of E10.5 embryos from the *Csf1^{Cre} Trim71* cKO mouse line, showing that *Csf1^{Cre} Trim71* cKO does not cause embryonic lethality or obvious morphological defects during embryonic development (Fig S2D, E, described on p. 9).

Q1.8: *Why is Trim71 expression in the yolk sac so punctate (supp fig 4C), as compared to the other tissues (supp fig 4B, D)?*

A1.8: We have validated the specificity of the Trim71 antibody in the yolk sac by the absence of signal following staining of *Trim71* *-/-* yolk sacs and embryos. To this end, we stained *Trim71* *+/+* and *Trim71* *-/-* E9.5 embryos and E11.5 yolk sacs with the Trim71 antibody used in this study. We again observed the dot-like Trim71 staining pattern in the *Trim71* *+/+* yolk sac and the diffuse staining pattern in the *Trim71* *+/+* embryo proper, whereas no staining at all was detected upon Trim71-KO. These data show the specificity of the observed staining patterns.

[Figure removed by editorial staff per authors' request]

The punctuate Trim71 expression in the yolk sac is somewhat similar to previous staining patterns in cell lines, where Trim71 colocalizes with P-body markers such as Ago2 (Torres-Fernández et al. 2021). Trim71 might thus localize to P-bodies within cells of the yolk sac based on the observed staining pattern, but this is a speculation and we currently do not have any further evidence supporting this hypothesis, and therefore did not detail this in the manuscript text. See also A2.4 and A3.9.

Q1.9: Why are there large differences in some cell populations (supp Fig 5C)? e.g. primitive streak or ExE

A1.9: The differences in these cell populations observed in the scRNA-seq dataset of E7.5 embryos might indicate that loss of *Trim71* influences their overall number, which needs to be validated by a more robust method of quantification, e.g. by immunofluorescence staining, in future experiments. We updated the results and discussion sections to now address changes in cell numbers observed in the E7.5 scRNA seq experiment and added a brief discussion of a possible role of Trim71 in the ExE endoderm (p. 12, p. 19), as well as an analysis of DEGs in the ExE endoderm (Fig S3F, Fig S5E).

Reviewer #2

In this paper, the authors investigate the function of the RNA-binding protein Trim71 during early murine embryogenesis. This paper is very well-written and the data are logically laid out and presented. The author show that Trim71 knockout embryos die at E10.5-E11.5 with significant defects of the cardiovascular system, including a mark reduction of primitive erythroid cells, vascular remodeling defects in the yolk sac, and a decrease in cardiac function (contractions and heart rate). These defects are in addition to the previously described neuronal and germ cell defects, and establish the likely cause of lethality in embryos lacking Trim71.

Consistent with defects in the cardiovascular system, markedly reduced migration of macrophage cells was noted into the head and remainder of the embryo body. Normal numbers of macrophage lineage cells and "EMP" were present in the yolk sac at E9.5 and E10.5 conceptuses. Consistent with these findings, functional deletion of Trim71 in erythro-myeloid progenitors/macrophage lineage cells using the inducible Csf1rCre mouse model failed to elicit defects in those hematopoietic lineages. Single cell RNA-Seq studies of E9.5 yolk sacs in wild-type and Trim71-null embryos confirmed the marked reduction in primitive erythroid cells and revealed gene expression differences in endothelial cells.

Since hematopoietic and (some, but not all) endothelial cells are thought to arise from common progenitors, Tie2Cre was used to ablate Trim71 from these cell types. Surprisingly, there were only mild endothelial defects and no evidence of functional cardiac or primitive erythroid defects in Trim71-targeted embryos. Given these results, indicating that Trim71 is not directly responsible for the hematopoietic and cardiac defects noted in the KO embryo, the investigators postulated that Trim71 must be functioning earlier in development and examined gastrulating mouse embryos at E7.5 by scRNA-Seq.

Q2.1: Single cell RNA-Seq studies at E7.5 were conducted to better understand the etiology of the cardiovascular defects. Decreases in "mesenchyme" and "hemato-endothelial progenitors" were noted. While not specifically described, it is interesting that the number of extraembryonic endoderm cells is markedly expanded (sFig. 5C) and their position in UMAP space is different (Fig. 6A). Since extraembryonic endoderm has been described as impacting vascular and hematopoietic development in the yolk sac (e.g., Miura, 1969; Palis, 1995; Bielinska, 1996; Belaousoff, 1999), an analysis of gene differences in this tissue should be undertaken.

A2.1: We have added an analysis of gene expression in the ExE endoderm in E7.5 embryos and E9.5 yolk sacs to the manuscript (Fig S3F and Fig S5E), and considered a putative role of the ExE endoderm for the observed phenotypes of *Trim71*-KO embryos in the discussion section (p. 19).

Q2.2: The single cell RNA-Seq analysis of 'mesodermal progenitors' leads to the identification of Eomes as a potential target of Trim71. Elegant mechanistic studies confirm that Eomes is a direct target of Trim71 binding and the high expression of Eomes in Trim71-null mesoderm is consistent with the known repressive function of Trim71. The authors suggest that altered expression of Eomes contributes to the defects in cardiovascular development, i.e., the defects in yolk sac endothelial cells, in primitive erythropoiesis, and the cardiac defects. While not necessary, the paper would be strengthened by evidence that overexpression of Eomes leads to one or more of these defects in differentiating mES cells.

A2.2: A study by Pfeiffer et al. (2018) has explored the effects of different levels of Eomes expression in ESC for their differentiation into mesodermal cell fates (<https://pubmed.ncbi.nlm.nih.gov/29382828/>). We have added this point to the discussion of the manuscript (p. 19).

Q2.3: Finally, the discussion very nicely brings up multiple excellent points, including the importance of flow for vascular remodeling of the yolk sac and the importance of Eomes in the emergence of primitive erythroid cells in the yolk sac. A final issue should be considered: The expression of Trim71 shown in Supplemental Figure 4B-D raises some questions. It would appear from sFig. 4B, that Trim71 is most highly expressed in extraembryonic (visceral) endoderm. Is that correct? If so, was that high expression evident in the transcriptomic data from Pijuan-Sala, et al.? If so, those data should be shown and the possibility that extraembryonic endoderm impacting vascular and hematopoietic development in the yolk sac entertained/discussed (as mentioned above).

A2.3: The scRNA seq data from embryos at E7.5 by our study (Fig S5D) show that Trim71 mRNA is expressed at equally high levels comparing the extraembryonic endoderm and other embryonic tissues (such as mesoderm) at this developmental stage. We have repeated the immunofluorescence staining of E7.5 WT embryos with the Trim71 antibody, including a secondary antibody control, and noticed that the extraembryonic endoderm was prone to unspecific binding of the secondary antibody. This may have contributed to the high signal in this layer shown in Fig S4B of the initial manuscript version. The updated version of Fig S4B shows the new Trim71 staining and the secondary antibody control with indications for unspecific binding of the secondary antibody, which could not fully be abolished using our immunofluorescence staining protocol. Nevertheless, we have added volcano plots of DEGs in the ExE endoderm (E7.5 whole embryo Fig S5E and E9.5 yolk sac Fig S3F) and discussed the potential role of this tissue in the contribution to yolk sac vascular development (p. 19). See also A3.8.

Q2.4: Additionally, in sFig. 4C, it is not clear to me that Trim71 is expressed in CD31+ cells at 9.5. The small green 'dashes' of positivity seems very different from the diffuse staining in sfigures B and D.

A2.4: We have performed an experiment to validate the specificity of the observed dot-like staining pattern (see A1.8). In the updated results text, we now emphasize the difference in staining pattern between yolk sac and embryo proper. To address if Trim71 is expressed within the EC in E9.5 yolk sacs, we performed a three-dimensional reconstruction of z stack images using Imaris. This analysis clearly shows Trim71 signal within the CD31+ EC (indication by arrows). We also detect dot-like Trim71 expression outside of the CD31+ EC, presumably from ExE endoderm cells. We now mention in the manuscript that the dot-like Trim71 expression is also present in other cells besides EC in the yolk sac (p. 10–11). See also A1.8 and A3.9.

[Figure removed by editorial staff per authors' request]

Reviewer #3:

1. A short summary of the paper, including description of the advance offered to the field.

In this study, Beckroge et al. demonstrate that disrupting Trim71 function during mouse embryogenesis leads to cardiovascular defects, which likely underpins the embryonic lethality observed in mutant embryos midway through gestation. The researchers identify noticeable defects beginning at embryonic day 9.5 (E9.5), including pronounced developmental delays in Trim71 knockout (KO) embryos. These delays are associated with vascular remodelling defects, the absence of major vitelline vessels in the yolk sac, a reduction in primitive erythrocytes, and impaired cardiac function. To pinpoint the timing of Trim71's role in yolk sac hematopoiesis and vascular development, they specifically disrupted Trim71 in hematoendothelial progenitors (using Tie2-Cre) and in erythro-myeloid progenitors (using Csf1r-Cre). The conditional knockout studies revealed that Trim71 expression in these progenitors is not the primary cause of the disruptions observed in Trim71 KO embryos. Through single-cell RNA sequencing of E7.5 Trim71 KO embryos, the team identified upregulation of the transcription factor Eomes in mesoderm and endoderm cells. Further investigation using an embryonic stem cell model of mesoderm formation showed that Trim71 directly regulates Eomes expression, as demonstrated by immunoprecipitation experiments. Collectively, this work establishes Trim71 as a key regulator of Eomes mRNA expression during early stages of murine gastrulation and underscores its crucial role in early yolk sac and cardiac development in the mouse embryo.

2. For each main point of the paper, please indicate if the data are strongly supportive. If not, explicitly state the additional experiments essential to support the claims made and the timeframe that these would require.

Main point #1: Trim71 is required for primitive erythropoiesis and cardiovascular development

The morphological analyses of developing mutant embryos via flow cytometry and fluorescence imaging strongly support the conclusions that Trim71 is required for normal cardiovascular development and primitive erythropoiesis.

Main point #2: Trim71-dependent control of gene expression during gastrulation determines proper vascular development and primitive erythropoiesis

Q3.1: The researchers demonstrate that Trim71 plays a critical role in yolk sac (YS) vascular development, primitive erythropoiesis, and heart development. They suggest that the observed defects likely stem from disruptions in Trim71-dependent gene expression during early mesoderm formation, particularly through its regulation of Eomes expression. Given that Trim71 is broadly expressed in various early embryonic cell types beyond the mesoderm, and considering that conditional disruption of Trim71 in the epiblast (for example, using a Sox2-Cre mouse line or by generating chimeras with Trim71 KO embryonic stem cells) has not been performed, the authors should acknowledge the possibility that these defects might arise from non-cell-autonomous roles of Trim71, such as in the visceral endoderm, which, along with the extraembryonic mesoderm, forms the visceral yolk sac.

A3.1: We have now acknowledged a possible cell-autonomous role of Trim71 in the ExE endoderm as the basis for yolk sac vascular phenotypes in the discussion. To underscore this, we added a citation showing that deletion of the transcription factor YY1 in the ExE endoderm leads to such phenotypes (Rhee et al. 2013). We have also added a paragraph to the discussion which refers to potentially testing the importance of the ExE endoderm for the yolk sac phenotypes observed in *Trim71*-KO embryos to the using a Sox2 Cre line, or with chimeras from Trim71-deficient mESC (p. 19).

Q3.2: While additional conditional knockout experiments could provide further insights, they are not strictly necessary, though they would be valuable. Instead, the authors should emphasize the number of differentially expressed genes identified in other cell types in their single-cell RNA sequencing (scRNA-seq) analyses at E7.5/E9.5. Currently, the focus is primarily on the endothelium and mesoderm, while the rationale for this focus is obvious, the authors should explore and outline gene expression changes in other cell types, such as the visceral endoderm. These additional analyses could help rule out non-cell-autonomous roles of Trim71 that might contribute to YS formation defects. For instance, if a significant number of differentially expressed genes are found in the visceral endoderm at E9.5, these changes could have induced alterations in gene expression within the Trim71 mutant endothelium, potentially underlying the observed vascular defects in the YS.

A3.2: We have added volcano plots of DEGs in the ExE endoderm to the figures (Fig S3F and S5E, respectively), and discussed the possibility of transcriptional changes in this tissue as a factor influencing the yolk sac development in *Trim71*-KO embryos (p. 19).

Q3.3: The Trim71 KO embryos exhibit developmental delays, which adds another layer of complexity to the interpretation of the data. It would be valuable for the researchers to integrate their findings with the scRNA-seq data presented in Imaz-Rosshandler (2024) from E8.5 to E9.5 YS sections. This integration could help assess whether the mutant endothelium aligns with a delayed differentiation trajectory, potentially explaining many of the differentially expressed genes identified in their study. Understanding whether the observed gene expression changes are a consequence of delayed development, rather than direct effects of Trim71 loss, could provide important insights into the mechanisms underlying the vascular defects in Trim71 KO embryos.

A3.3: To address this suggestion, we have extracted single cell transcriptomes from HEP and EC populations at E7.75, E8.0, E8.25 and E8.5 from the dataset reported by Pijuan-Sala et al. (2019), published by the same research group as the paper by Imaz-Rosshandler et al. (2024). We re-calculated a UMAP for these reference cells and projected the transcriptomes of *Trim71*-WT and *Trim71*-KO EC onto this UMAP using the functions FindTransferAnchors, IntegrateEmbeddings and ProjectUMAP from the Seurat package. The projections of EC do not show clear differences between the genotypes, with the majority of query cells from both genotypes locating to reference E8.5 EC at the same position in the UMAP (data below). By this preliminary analysis, we do not find clear evidence supporting the hypothesis that *Trim71*-KO yolk sacs have delayed endothelial cell differentiation.

[Figure removed by editorial staff per authors' request]

Q3.4: Finally, it is worth noting that Eomes is not only crucial for the formation of primitive erythrocytes but also for the development of cardiac mesoderm and heart formation, as shown by Costello et al. (2011). Given that Trim71 KO embryos exhibit cardiac function defects, the authors should reference this study to strengthen their discussion.

A3.4: We have added the respective reference to the discussion section (p. 19).

3. Lastly, indicate any additional issues you feel should be addressed (text changes, data presentation, statistics etc.).

Q3.5: Figure 3

Which markers were used to determine EMP (CD41, cKit, AA4.1?) and pMac percentages is not clearly presented in the text or figure. While this information is provided in the methods section, it would be beneficial to also include it in the main text for clarity and better accessibility.

A3.5: We defined yolk sac EMP as CD45^{low} Kit⁺ AA4.1⁺ based on the studies of Gomez-Perdiguerro et al. (2015, <https://pubmed.ncbi.nlm.nih.gov/25470051/>) and Iturri et al. (2021, <https://pubmed.ncbi.nlm.nih.gov/34062116/>). pMac were defined as CD45⁺ Kit⁻ CD11b⁺ based on Mass et al. (2016, <https://pubmed.ncbi.nlm.nih.gov/27492475/>). We added the surface markers used to define cell types by flow cytometry (EryP, EC, EMP, pMac, Macrophages) to the results section of the main text (p.6-8). The marker combinations for cell type identification are also described in the method section (p. 22).

Q3.6: Figure 4

The scRNA-seq analysis of the yolk sac at E9.5 should include a comparison of the molecular properties of the cells with the gastrulation atlas (Imaz-Rosshandler 2024) to investigate whether the observed transcriptional disruptions are linked to a developmental delay. Specifically, this comparison could reveal if the endothelium in the Trim71 KO embryos is less mature, mapping to earlier developmental stages, such as E8.5. The decreased expression of Cdh5 observed in the data supports this possibility. Furthermore, many of the DEGs might be due to the block in maturation.

A3.6: A preliminary comparison of the EC transcriptome from E9.5 *Trim71*-WT and *Trim71*-KO yolk sacs with the gastrulation atlas is shown in A3.3. Of note, the current literature suggests that *Trim71* functions as an inhibitor of differentiation. As such, *Trim71*-KO across

multiple cell types, including mESC, epidermal cells and neural progenitor cells, results in accelerated expression of differentiation markers (Mitschka et al. 2015; Slack et al. 2000; Chen et al. 2012). It is, however, of great interest to us to analyze alterations in differentiation kinetics upon loss of Trim71 comparing cells types derived from different germ layers (e.g., mesodermal-derived and ectodermal-derived), and we plan to address germ-layer specific effects of *Trim71*-KO in future studies.

Q3.7: What about the transcriptional changes in other cell types beyond the endothelium? Given that Trim71 is broadly expressed, it may also play a role in the extraembryonic endoderm lineage, which is important for primitive erythropoiesis. This could potentially explain some of the defects observed in the mutants. The authors should at least address this possibility in the discussion. Additionally, it would be helpful to plot differentially expressed genes (DEGs) for each cell type, including primitive erythrocytes, to provide a more comprehensive view of Trim71's impact across various lineages. This should be included in supplement.

A3.7: We have analyzed transcriptional changes between genotypes in all cell types from both scRNA seq datasets (E7.5 whole embryos, E9.5 yolk sac). The resulting DEG for each cell type have been added to the supplementary information of the manuscript in the form of excel tables that can easily be explored by the reader. The DEGs detected in EryP and ExE endoderm (E7.5 and E9.5) were plotted as volcano plot and were added to the manuscript (Fig S3E, F and Fig S5E). Plotting the DEGs for all cell types from both scRNA-seq experiments would have produced a total of 19 graphs, which we thought would add too much unnecessary detail to the study.

Q3.8 SFig4

- Is there strong expression in the visceral endoderm?

- It would be helpful to include a control or at least indicate what is considered a positive signal in the images, perhaps using arrowheads for clarity.

A3.8 There is Trim71 expression in the visceral endoderm, but at least at the mRNA level the expression is equally high as compared to other tissues at E7.5 (see Fig S5D). We performed another immunofluorescence staining of Trim71 in wildtype E7.5 embryos, and included a secondary antibody control staining (only added secondary antibody but omitted the Trim71 antibody). This staining showed that the ExE endoderm tissue is prone to unspecific binding of the secondary antibody, which was the reason for the strong signal in this tissue in the image shown in the initial version of the manuscript. In the updated manuscript version, we show the new image of the Trim71 staining alongside with the secondary antibody control staining and indicate areas of background staining with stars (Fig S4B). See also A2.3.

Q3.9: Additionally, the staining pattern in the E9.5 yolk sac appears noticeably different from that in the gastrula and embryo. It would be valuable to address this discrepancy and provide an explanation for the observed differences.

A3.9: We have added a sentence to the results section to address this point: "Of note, while the Trim71 protein signal was diffuse cytoplasmic in E7.5 embryos and CD31+ cells of the dorsal aorta, Trim71 protein expression in the yolk sac was restricted to discrete foci within CD31+ EC and also in CD31- cells, reminiscent to previously observed Trim71 dot-like staining patterns in cell lines (Torres-Fernández et al. 2021)." (p. 10–11). See also A1.8 and A2.4.

Q3.10: SFig5- A, include the genotype label in the heatmap

A3.10: The heatmap is composed of the merged dataset comprising both genotypes, therefore no genotype labels can be added to this representation.

Q3.11: Figure 5: Would be nice to see the number of DEGs per cell type, to help with interpretation of what underlies the defects in YS hematopoiesis

A3.11: We have added excel files with all identified DEGs by cell type from both scRNA seq experiments to the supplementary information.

Q3.12: Should distinguish definitive endoderm in the labels of the cell types from the visceral endoderm (ExE)

A3.12: We have changed the labeling of endoderm to definitive endoderm (Fig 3A, H, I and Fig S5A–D).

Q3.13: In the discussion, the authors propose that the overexpression of *Eomes* in the mesoderm of *Trim71* KO embryos might contribute to the observed reduction in primitive erythropoiesis. Notably, a recent preprint study (<https://www.biorxiv.org/content/10.1101/2024.08.13.607790v1>) demonstrates that *Eomes* knockout blocks yolk sac formation both *in vitro* and *in vivo*, suggesting that the reduction in primitive erythropoiesis observed in Harland et al., 2021, may be due to an earlier loss of yolk sac-fated mesoderm. Additionally, the discussion should reference further studies that have shown *Eomes* induction in embryonic stem cell (ESC) models, in various signaling contexts and at different doses, promotes the formation of lineages where *Eomes* is normally required, such as definitive endoderm and cardiac mesoderm (<https://www.ncbi.nlm.nih.gov/pmc/articles/PMC3321156/>, <https://www.nature.com/articles/s41467-017-02812-6>). These references would provide a broader context for understanding the role of *Eomes* in lineage specification.

A3.13: The mentioned references were very helpful to more precisely understand the potential effect of changed *Eomes* expression in the *Trim71*-KO embryos and were added to the discussion section (p. 18–19).

Q3.14: Figure 7

- What cell types are being generated in the embryoid bodies (EBs)-endoderm, mesoderm?

It would have been informative to stain for *Flk1/PdgfrA* and assess the formation of primitive erythrocytes (EryP) later in differentiation, using the flow cytometry markers mentioned in the paper. Since *Eomes* is expressed in the anterior primitive streak during the formation of definitive endoderm and cardiac mesoderm, it would be valuable to clarify the specific contexts in which these critical molecular experiments were conducted.

A3.14: Our serum-containing differentiation protocol is based on a protocol by Pearson et al. (2008), who used this protocol to generate immunophenotypic *Flk1+* cells after 3 days of culture within embryoid bodies. We used serum supplemented differentiation medium instead of serum-free medium supplemented with growth factors because in our hands, mESC did not survive differentiation under serum-free conditions. While our used differentiation model is sufficient to generate mesodermal cells, we acknowledge that the presence of serum in the differentiation medium could also lead to the induction of non-mesodermal cell fates (e.g., endoderm). We have therefore changed the sentence results section of the manuscript from

“Cells were differentiated within embryoid bodies towards the mesodermal lineage for four days”

to

“Cells were differentiated as embryoid bodies in the presence of serum for four days by the removal of leukemia inhibitory factor (LIF) and 2i (CHIR99021 and PD0325901) from the medium” (p. 13).

To characterize the cell types generated in our differentiation protocol we performed qPCR for *Flk1* and *Pdgfra*. *Flk1* expression was induced by differentiation of *Trim71*-flox but not in *Trim71*-KO mESC, that loss of *Trim71* interferes with normal mesodermal differentiation in this *in vitro* model. *Pdgfra* mRNA was not induced by differentiation in either genotype. The figures were added to the manuscript in Fig S6A, B and are described in the results section (p. 13).

We would like to emphasize that the main objective of the used mESC model in this study was to analyze the direct molecular interaction between *Trim71* and *Eomes* mRNA, which we

were able to achieve with the given differentiation protocol. In the future we plan to further define the role of Trim71 in mesodermal differentiation using a strictly defined serum-free mESC differentiation protocol towards to induce mesodermal cell generation.

Q3.15: Why do you think Lhx1 expression remains unchanged, despite it being a direct Eomes target? Is it possible that you examined it too early in the developmental process?

A3.15: We expect that effects of increased Eomes expression on Lhx1 induction could become apparent with extended differentiation of mESC for longer than day 4, at which time point we observed the increased Eomes expression in *Trim71*-KO mESC. If Eomes expression is only beginning to be increased at day 4, it might have not been long enough to observe secondary effects caused by the increase in Eomes expression. Nevertheless, the chosen analysis timepoint at day 4 of differentiation was adequate to study a potential direct regulation of Lhx1 mRNA levels by Trim71, which we could exclude by the conducted experiments.

Reviewer #4:

In this study, Beckröge and colleagues revealed that Trim71 KO embryos show deficiency in primitive erythropoiesis, yolk sac vasculature, heart function and circulation. Trim71 deletion cause transcriptional change in mRNA processing and RNA splicing related genes in the mesoderm at E7.5. Interestingly, Trim71 KO show elevated Eomes expression not only in the mesoderm, but also endoderm. This involves the direct interaction between Trim71 and Eomes mRNA. While these findings are of interest, there is additional experiments needed to confirm some of findings.

Main comments

Q4.1: Fig. 1A: It is unclear from the bright field images if the formation of the primitive streak in Trim71 KO embryos is impaired or not. Sectioning and H&E staining or whole mount staining for Brachyury/T or other primitive streak marker gene will be required to show if primitive streak formation is normal in Trim71 deletion.

A4.1: We have performed H&E staining of sections from E7.5 *Trim71* *+/+* and *Trim71* *-/-* embryos (shown below). While these stainings showed no differences in morphology between the genotypes, they were not sufficient to clarify the normal formation of a primitive streak. We have therefore modified the text in the results part to omit conclusions on the proper formation of the primitive streak in *Trim71*-KO embryos and removed the labeling of the primitive streak region from the bright field images in Fig 1A. We would, however, like to emphasize that mesodermal cells appear to be produced in normal numbers in *Trim71*-KO embryos (Fig S5C).

[Figure removed by editorial staff per authors' request]

Q4.2: Fig. 1A and B: The authors mentioned that Trim71 KO embryos appear pale in colour from E9.5 onwards, but this was not observed in a previous study (Mitschka et al., 2015). Is this due to using a different genetic background of Trim71 mutants, although the authors used the same mouse lines as Mitschka et al? What is the difference from Mitschka et al?

A4.2: The *Trim71* mutant mouse line analyzed in this study is the same as in Mitschka et al. (2015), and has been maintained on a C57BL/6J background. The phenotype was not noticed during the work for the respective previous study, because the research interest of the group was primarily focused on the neural tube phenotype at the time.

Q4.3: Supplemental Fig. S1D-F: The quantification method for the relative number of ECs is unclear. The methods section defines ECs as CD31+/AA4.1- cells, yet Ter119- cells are counted as ECs. This discrepancy should be clarified in the methods section.

A4.3: In the quantification of the flow cytometry data, the definition of EC is not only being negative for Ter119, but in addition to this marker as CD31+ AA4.1- in the yolk sac (to exclude EMP, that also express CD31 but are AA4.1+) and as CD31+ in the embryo head and embryo body. The graphs of cell number quantifications by flow cytometry display relative abundances of the cell population either as percentage of all live cells (used for quantification of EryP), or as percentage of all non-erythroid Ter119- cells (for quantification of all other cell populations). We chose to display the data in this way, rather than showing absolute cell counts, to account for the overall decrease in cellularity of *Trim71*-KO embryos. To avoid confusion for the reader, we changed the labeling from *EC [% of Ter119- cells]* to *EC [as % of Ter119- cells]* in all figures. We changed this labeling also for all other quantified cell populations and added a description of the quantification to the methods section (p. 22).

Q4.4: Fig 2: The manuscript does not clearly address whether heart development in Trim71 KO embryos is normal. The authors should investigate if the reduced heart beat rate is linked to developmental defects in the heart, and whether cardiac abnormalities and circulatory issues could contribute to embryonic lethality.

A4.4: In the time frame of the revision for this publication we were not able to adequately investigate if *Trim71*-KO embryos have an impaired cardiac development, although their decrease in heart function clearly indicates the presence of such a phenotype. We thank the reviewer for this suggestion that we want to include in our future studies. We have considered the contribution of the impaired circulation to the embryonic lethality of *Trim71*-KO embryos in the text of the discussion section. It is known that embryonic lethality can arise from defects originating in impaired heart contractility (Huang et al. 2003), but considering the shared mesodermal origin of cardiomyocytes, endothelial cells and blood cells, and the deficiencies of *Trim71*-KO embryos in tissues derived from all of these cell types, we did not pinpoint the lethality of these embryos to just one of these processes. For this reason, we chose to phrase the respective text of the results section in the following way (p. 16):

“Considering that defects in either component of the circulatory system are sufficient to drive embryonic lethality, the combined defects in all major parts of the circulatory system of *Trim71*-KO embryos provide an adequate explanation for their developmental arrest and lethality.”

Q4.5: Fig. 4A: The population for EC looks small in the scRNA-seq data. Is this consistent with the flow data for EC population in the yolk sac of E9.5?

A4.5: In the scRNA seq data of E9.5 yolk sacs, the EC population ranges from 0.8% to 3% of all cells, while our flow cytometry data, EC make up on average 13% of all live cells in E9.5 yolk sacs (in Fig. S1D, EC are shown as relative percentage of non-erythroid Ter119-negative cells, which is why these values do not correspond to the percentages of the scRNA-seq data, which are given as percent of total cells). EC numbers in the yolk sac are thus within the same order of magnitude comparing scRNA seq and flow cytometry data from our study.

Q4.6: Regarding scRNA-seq for E7.5 embryos, one whole embryo of each genotype is not sufficient to detect biological effects of Trim71 deletion. Given the variability and different developmental speed within the same litters, more replicates (at least two, preferably more than three) are required to accurately represent gene expression profiles in Trim71 KO embryos.

A4.6: Analyzing three or more embryos per genotype for the scRNA-seq experiments was in essence not feasible due to the tremendously high costs of such additional experiments. The hypothesis that we formulated based on the results of the E7.5 embryo scRNA-seq

experiment shown in the study, namely the regulation of Eomes mRNA expression by Trim71, was validated using the mESC in vitro model. In our view, the observation of this phenotype in two unrelated models, together with the detection of a Trim71 responsive element in the Eomes 3'UTR and the CLIP data, provides ample support of the hypothesis that Eomes is regulated by Trim71.

Q4.7: Fig. 7A and B: Are the differentiation kinetics, such as the emergence of Flk1 high cells into mesodermal lineages, equivalent in Trim71 KO compared to WT?

A4.7: Analysis of Flk1 mRNA expression by qPCR showed decreased induction of Flk1 expression upon differentiation of *Trim71*-KO mESC compared to WT mESC. We have added the graphs to Fig S6A, B and described them in the results section (p. 13).

Q4.8: Does Trim71 R595H ESC show upregulation of Eomes mRNA when differentiated to embryoid bodies and mesoderm lineages for 4 days, similar to Trim71 KO? Also, does Trim71 R595H mutant ESC show defects in mesoderm and erythroid differentiation? It is unclear if the misinteraction with Eomes in Trim71 deletion causes erythroid differentiation defects in the mutant embryos.

A4.8: We performed qPCR analysis of Eomes expression at day 4 of differentiation in *Trim71*-flox and *Trim71*-R595H cells. Eomes mRNA levels were tendentially increased in the R595H mutant cells, we added the graph to Fig S6C and described in the results section (p. 14). Performing differentiation assays of mESC upon Trim71 mutation was beyond the scope of our manuscript, but we modified the discussion section to emphasize that we do not claim that upregulation of Eomes represents the only mechanisms from which the developmental defects of *Trim71*-KO embryos could arise.

Minor comments

Q4.9: Page 5: "characteristic embryonic morphology until E9.5" - As E9.5 Trim71 KO embryos show size defects and cranial neural tube closure defects, Trim71 KO embryos acquired the characteristic morphology until E8.5 and showed defects afterwards.

A4.9: Changed text to (p. 5–6):

"Trim71-KO embryos were morphologically indistinguishable from wildtype littermate control (WT) embryos at E7.5 and E8.5 (Fig 1A). In agreement with previous observations, Trim71-KO embryos were smaller than WT embryos at E9.5, and displayed a cranial neural tube closure defect (Fig 1A)"

Q4.10: Fig. 1C: EryP is analyzed by Ter119 and CD71 expression by flow cytometry in many other studies. Can the author examine CD71 /Ter119 in the E10.5 yolk sac?

A4.10: We have analyzed the median fluorescence intensity of CD71 surface expression on yolk sac EryP at E9.5 and E10.5. There was no difference in CD71 levels between *Trim71*-WT and *Trim71*-KO EryP. These data have been added to Fig S1E and are described on p. 6. We have also added a representative flow cytometry plot to Fig S1D showing Ter119 and CD71 in yolk sac EryP overlaid on all live cells, clearly demonstrating that the EryP population identified by our gating strategy (Ter119⁺ CD45⁺) expresses CD71.

Q4.11: Supplementary Fig. 2B and C: Does MFI mean "Mean Fluorescence Intensity"?

A4.11: MFI refers to median fluorescence intensity. With the addition of new flow cytometry data to Fig S1D, we now explain the abbreviation in the figure legend of Fig S1 and also in Fig 7G.

Q4.12: Fig. 3 and supplemental Fig. S1D-F: All cell types are shown as "% of Ter119- cells". Please correct this if these are mislabeled.

A4.12: Throughout the manuscript, all non-erythroid cell populations are quantified as relative percentage of non-erythroid cells to account for the overall decrease in cellularity and EryP of *Trim71*-KO embryos. The labeling was intentional, but we have now changed it to [**as** % of *Ter119*- cells] for all figures to make it clearer for the reader. See also A4.3.

Q4.13: Fig. 6B: The use of light gray and dark gray in the UMAP plot makes it hard to see the difference between WT and Trim71 KO cells. A more distinct colour combination is recommended to better illustrate the distribution changes of Trim71 KO cells.

A4.13: We have changed the colours to red and blue in both figures of scRNA-seq experiments (Fig 4B and Fig 6B).

Q4.14: Is Trim71 deletion by Tie2 Cre in conditional knockout embryos confirmed by qPCR or IF? What was the efficiency of Trim71 deletion in the conditional knockout?

A4.14: We quantified *Trim71* expression in the *Tie2^{Cre} Trim71* cKO line by immunofluorescence staining of *Trim71* and *CD31* in transversal sections of *Tie2* +/+ and *Trim71* fl/fl *Tie2* Cre/+ embryos at E9.5. The graph below shows representative images and the quantification of the mean *Trim71* signal intensity in >10 *CD31*⁺ cells from n=3 embryos per genotype. These data show that *Trim71* is reliably targeted in EC, but not surrounding *CD31*⁻ cells, in the *Tie2^{Cre} Trim71* cKO line.

[Figure removed by editorial staff per authors' request]

Q4.15: The details of antibodies used in this study should be listed.

A4.15: We have added the antibody details to the method section (p. 21–23, p. 25).

References

- Biben, C.; Weber, T. S.; Potts, K. S.; Choi, J.; Miles, D. C.; Carmagnac, A. et al. (2023): In vivo clonal tracking reveals evidence of haemangioblast and haematomesoblast contribution to yolk sac haematopoiesis. In: *Nature communications* 14 (1), S. 41. DOI: 10.1038/s41467-022-35744-x.
- Chen, Jianfu; Lai, Fan; Niswander, Lee (2012): The ubiquitin ligase mLin41 temporally promotes neural progenitor cell maintenance through FGF signaling. In: *Genes & development* 26 (8), S. 803–815. DOI: 10.1101/gad.187641.112.
- Huang, Chengqun; Sheikh, Farah; Hollander, Melinda; Cai, Chengleng; Becker, David; Chu, Po-Hsien et al. (2003): Embryonic atrial function is essential for mouse embryogenesis, cardiac morphogenesis and angiogenesis. In: *Development (Cambridge, England)* 130 (24), S. 6111–6119. DOI: 10.1242/dev.00831.
- Imaz-Rosshandler, Ivan; Rode, Christina; Guibentif, Carolina; Harland, Luke T. G.; Ton, Mai-Linh N.; Dhapola, Parashar et al. (2024): Tracking early mammalian organogenesis - prediction and validation of differentiation trajectories at whole organism scale. In: *Development (Cambridge, England)* 151 (3). DOI: 10.1242/dev.201867.
- Mitschka, Sibylle; Ulas, Thomas; Goller, Tobias; Schneider, Karin; Egert, Angela; Mertens, Jérôme et al. (2015): Co-existence of intact stemness and priming of neural differentiation programs in mES cells lacking Trim71. In: *Scientific reports* 5, S. 11126. DOI: 10.1038/srep11126.
- Pearson, Stella; Sroczynska, Patrycja; Lacaud, Georges; Kouskoff, Valerie (2008): The stepwise specification of embryonic stem cells to hematopoietic fate is driven by sequential exposure to Bmp4, activin A, bFGF and VEGF. In: *Development (Cambridge, England)* 135 (8), S. 1525–1535. DOI: 10.1242/dev.011767.
- Pijuan-Sala, Blanca; Griffiths, Jonathan A.; Guibentif, Carolina; Hiscock, Tom W.; Jawaid, Wajid; Calero-Nieto, Fernando J. et al. (2019): A single-cell molecular map of mouse gastrulation and early organogenesis. In: *Nature* 566 (7745), S. 490–495. DOI: 10.1038/s41586-019-0933-9.
- Rhee, Siyeon; Guerrero-Zayas, Mara-Isel; Wallingford, Mary C.; Ortiz-Pineda, Pablo; Mager, Jesse; Tremblay, Kimberly D. (2013): Visceral endoderm expression of Yin-Yang1 (YY1) is required for VEGFA maintenance and yolk sac development. In: *PloS one* 8 (3), e58828. DOI: 10.1371/journal.pone.0058828.
- Slack, F. J.; Basson, M.; Liu, Z.; Ambros, V.; Horvitz, H. R.; Ruvkun, G. (2000): The lin-41 RBCC gene acts in the *C. elegans* heterochronic pathway between the let-7 regulatory RNA and the LIN-29 transcription factor. In: *Molecular cell* 5 (4), S. 659–669. DOI: 10.1016/S1097-2765(00)80245-2.
- Torres-Fernández, Lucia A.; Emich, Jana; Port, Yasmine; Mitschka, Sibylle; Wöste, Marius; Schneider, Simon et al. (2021): TRIM71 Deficiency Causes Germ Cell Loss During Mouse Embryogenesis and Is Associated With Human Male Infertility. In: *Frontiers in cell and developmental biology* 9, S. 658966. DOI: 10.3389/fcell.2021.658966.

January 21, 2025

RE: Life Science Alliance Manuscript #LSA-2024-02956-TR

Prof. Waldemar Kolanus
University of Bonn
LIMES Program Unit Molecular Immune and Cell Biology, Laboratory of Molecular Immunology
Carl-Troll-Straße 31, Bonn
Bonn, NRW 53115
Germany

Dear Dr. Kolanus,

Thank you for submitting your revised manuscript entitled "Impaired primitive erythropoiesis and defective vascular development in Trim71-KO embryos". We would be happy to publish your paper in Life Science Alliance pending final revisions necessary to meet our formatting guidelines.

- please be sure that the authorship listing and order is correct
- please add the Twitter and Bluesky handles of your host institute/organization as well as your own or/and one of the authors in our system
- please note that the titles in the system and manuscript file must match
- please add a Conflict of Interest statement to your main manuscript text
- please add your main, supplementary figure, and movie legends to the main manuscript text after the references section and remove the SI file with this info
- please indicate in the Materials and Methods section that approval was granted to conduct the mouse work, and who granted that approval

LSA now encourages authors to provide a 30-60 second video where the study is briefly explained. We will use these videos on social media to promote the published paper and the presenting author (for examples, see <https://docs.google.com/document/d/1-UWCfbE4pGcDdcgzcmiuJI2XMBJnxKYeqRvLLrLS08s/edit?usp=sharing>). Corresponding or first-authors are welcome to submit the video. Please submit only one video per manuscript. The video can be emailed to contact@life-science-alliance.org

A. FINAL FILES:

B. MANUSCRIPT ORGANIZATION AND FORMATTING:

Sincerely,

January 27, 2025

RE: Life Science Alliance Manuscript #LSA-2024-02956-TRR

Prof. Waldemar Kolanus
University of Bonn
LIMES Program Unit Molecular Immune and Cell Biology, Laboratory of Molecular Immunology
Carl-Troll-Straße 31, Bonn
Bonn, NRW 53115
Germany

Dear Dr. Kolanus,

Thank you for submitting your Research Article entitled "Impaired primitive erythropoiesis and defective vascular development in Trim71-KO embryos". It is a pleasure to let you know that your manuscript is now accepted for publication in Life Science Alliance. Congratulations on this interesting work.

DISTRIBUTION OF MATERIALS:

Again, congratulations on a very nice paper. I hope you found the review process to be constructive and are pleased with how the manuscript was handled editorially. We look forward to future exciting submissions from your lab.

Sincerely,
